# Geometric Hyena Networks for Large-scale Equivariant Learning

Artem Moskalev [1]   Mangal Prakash [1]   Junjie Xu [1 2 3]   Tianyu Cui [1]   Rui Liao [1]   Tommaso Mansi [1]

## Abstract

Processing global geometric context while preserving equivariance is crucial when modeling biological, chemical, and physical systems. Yet, this is challenging due to the computational demands of equivariance and global context at scale. Standard methods such as equivariant self-attention suffer from quadratic complexity, while local methods such as distance-based message passing sacrifice global information. Inspired by the recent success of state-space and long-convolutional models, we introduce Geometric Hyena, the first equivariant long-convolutional model for geometric systems. Geometric Hyena captures global geometric context at sub-quadratic complexity while maintaining equivariance to rotations and translations. Evaluated on all-atom property prediction of large RNA molecules and full protein molecular dynamics, Geometric Hyena outperforms existing equivariant models while requiring significantly less memory and compute that equivariant self-attention. Notably, our model processes the geometric context of $30k$ tokens $20\times$ faster than the equivariant transformer and allows $72\times$ longer context within the same budget.

## 1. Introduction

Modeling global geometric context with equivariance is crucial in many real-world tasks. The properties of a protein depend on the global interaction of its residues (Baker & Sali, 2001). Similarly, the global geometry of RNA dictates their functional properties (Sato et al., 2021). In vision, modeling global geometric context is crucial when working with point clouds or meshes (De Haan et al., 2020). In all these tasks, maintaining equivariance while capturing global context is essential for robust modeling and prediction.

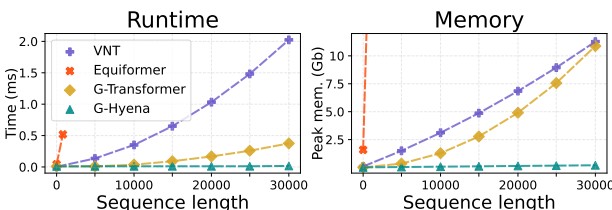

Figure 1. **Left:** GPU forward runtime comparison. Geometric Hyena scales sub-quadratically and achieves a considerable speedup compared to other equivariant models with global context. **Right:** Peak GPU memory consumption for G-Hyena is the most efficient for long sequences.

Processing global geometric context with equivariance is challenging due to the computational demands of processing high-dimensional data at scale. Existing methods either rely on global all-to-all operators such as self-attention (Liao & Smidt, 2023; de Haan et al., 2024; Brehmer et al., 2023), which scale poorly due to their quadratic complexity, or they restrict processing to local neighborhoods (Thomas et al., 2018; Köhler et al., 2020), losing valuable global information. This limitation is a significant practical bottleneck, necessitating more efficient solutions for scalable equivariant modeling with a global geometric context.

An efficient method for modeling global context should support easy parallelization during training while maintaining manageable computational costs at inference. One approach involves recurrent operators (Orvieto et al., 2023; De et al., 2024), which provide bounded compute but lack easy parallelization. Another family of methods relies on self-attention (Vaswani et al., 2017) allowing parallel processing at the cost of quadratic computational complexity. The most recent advances leverage state-space (Gu et al., 2021b; Fu et al., 2022; Gu & Dao, 2023) and long-convolutional (Romero et al., 2021; Poli et al., 2023) frameworks, enabling global context reasoning in sub-quadratic time with easy parallelization. Extending these models to accommodate equivariance remains an unexplored direction.

Inspired by the recent success of state-space and long-convolutional methods, we propose Geometric Hyena that efficiently models global geometric context in sub-quadratic time while preserving equivariance to rotations and translations. We focus on a subset of geometric graphs where

[1]Johnson and Johnson Innovative Medicine [2]The Pennsylvania State University [3]This work was done while the author was an intern at Johnson and Johnson. Correspondence to: Artem Moskalev <ammoskalevartem@gmail.com>.

*Proceedings of the 42nd International Conference on Machine Learning*, Vancouver, Canada. PMLR 267, 2025. Copyright 2025 by the author(s).

the canonical order can be established such as biomolecules. The focus on ordered geometric graphs differentiates Geometric Hyena from other equivariant frameworks and allows to leverage efficient sequence operators - long convolutions. For equivariance, we introduce vector long convolution that utilizes vector products between equivariant queries and keys. The vector long convolution is implemented in the Fourier domain, achieving $O(N \log N)$ computational complexity. We further show how to extend this vector long convolution to geometric long convolution by accommodating interaction between invariant and equivariant subspaces to model more complex geometric relations.

We conduct experiments on four real-world datasets spanning multiple large molecule property prediction and molecular dynamics simulation tasks. These include Open Vaccine (Das et al., 2020) and Ribonanza-2k (He et al., 2024) datasets for stability and degradation prediction where our model outperforms other state-of-the-art equivariant baselines by up to 15%. Moreover, it achieves a 6% improvement on the mRNA switching-factor prediction task (Groher et al., 2018) and outperforms other equivariant methods by 9% on all-atom protein molecular dynamics prediction. In addition, we introduce a new geometric associative recall task for mechanistic interpretability (Olsson et al., 2022) of geometric models. Experiments on this new geometric associative recall problem further validate the strong performance of our model and shed light on its scaling behavior. Overall, our results suggest that Geometric Hyena can outperform local and global equivariant methods while requiring less memory and compute for long geometric sequences. In particular, for a sequence of $30k$ tokens, Geometric Hyena runs $20\times$ faster than equivariant self-attention methods (see Figure 1). Notably, when the equivariant transformer runs out of memory on sequences over $37k$ tokens, *our model can handle up to $2.7M$ million tokens on a single A10G GPU*, providing up to $72\times$ longer context length within the same computational budget.

To sum up, we make the following contributions:

- We propose Geometric Hyena, the first equivariant long-convolutional architecture tailored for large geometric graphs with canonical ordering. Our model is designed to efficiently process global geometric context in sub-quadratic time.

- We show that Geometric Hyena outperforms local and global equivariant methods, and can surpass equivariant self-attention on large-scale molecular property prediction and all-atom protein MD prediction tasks while requiring significantly less memory and compute.

- We propose a new geometric associative recall task for the mechanistic interpretability of equivariant models on geometric data.

## 2. Geometric Hyena Network

The Geometric Hyena is designed for tasks that require modeling invariant and equivariant features in geometric graphs where canonical order can be established. The canonical ordering of a geometric graph implies a unique and unambiguous enumeration of its nodes. For instance, such canonical ordering is naturally established by IUPAC rules (Damhus et al., 2005) for biomolecules.

An ordered geometric graph of $N$ nodes can be written as a sequence $(\mathbf{x}_i, \mathbf{f}_i)_{i=1}^N = (\mathbf{x}_1, \mathbf{f}_1), \ldots, (\mathbf{x}_N, \mathbf{f}_N)$ where $\mathbf{x}_i \in \mathbb{R}^3$ represents vector features (e.g. coordinates or velocities), and $\mathbf{f}_i \in \mathbb{R}^d$ represents scalar features (e.g. atom types or fingerprints). We call $\mathbf{x}_i$ *geometric or vector token*, and $\mathbf{f}_i$ is a *scalar token*. In addition, a geometric graph can have edge attributes $e_{ij} \in \mathbb{R}^e$ between two nodes which are treated as scalar edge features. When working with geometric graphs, a neural network must respect symmetries of the input space and hence be equivariant with respect to geometric tokens and invariant with respect to scalar tokens.

Geometric Hyena consists of invariant and equivariant streams responsible for processing scalar and vector tokens respectively. Formally, the Geometric Hyena model $\Psi : (\mathbb{R}^3 \times \mathbb{R}^d)^N \to (\mathbb{R}^3 \times \mathbb{R}^d)^N$ satisfies equivariance property:

$$(L_g(\hat{\mathbf{x}}_i), \hat{\mathbf{f}}_i)_{i=1}^N = \Psi\left((L_g(\mathbf{x}_i), \mathbf{f}_i)_{i=1}^N\right) \qquad (1)$$

where $L_g : \mathbb{R}^3 \to \mathbb{R}^3$ is a representation of a group action $g \in SE(3)$. Thus, geometric tokens $\mathbf{x}_i$ transform accordingly with the group action while scalar tokens $\mathbf{f}_i$ remain invariant.

We make the overall information flow similar to the information flow of a transformer, as illustrated in Figure 2. The input is first mapped into keys, queries, and values via a projection layer. Next, the global context is aggregated via long convolution. Finally, the result of the long convolution is passed through the gating mechanism, and multiplied with values. We design each layer to be equivariant with respect to transformations of geometric tokens, and invariant for scalar tokens. This way, a model consisting of a composition of equivariant layers is equivariant (Weiler & Cesa, 2019). We provide formal proof of equivariance for each module of Geometric Hyena in Appendix A.4.2.

Next, we will detail key architectural components of Geometric Hyena as shown in Figure 2.

### 2.1. Equivariant projection layer

The equivariant projection layer $\phi : \mathbb{R}^3 \times \mathbb{R}^d \to \mathbb{R}^3 \times \mathbb{R}^d$ is a key component of our Geometric Hyena. It serves three primary functions: embedding scalar and vector tokens,

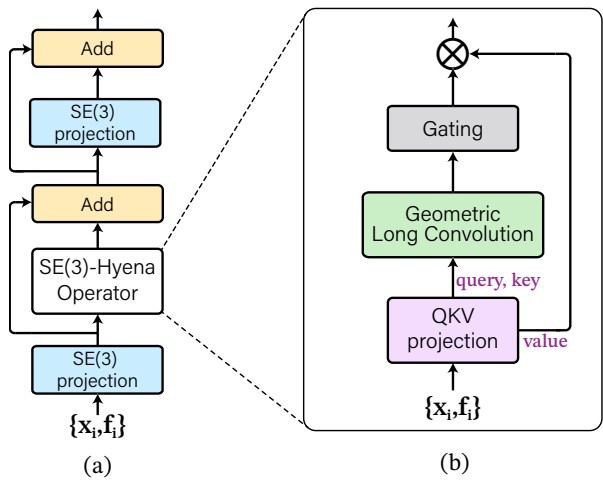

*Figure 2.* **Geometric Hyena block. (a)** Geometric Hyena block includes the SE(3)-Hyena operator and equivariant projections. **(b)** The SE(3)-Hyena operator includes query, key, value projection, geometric long convolution for global context aggregation, and gating.

mapping embedded tokens into query, key, and value triplets and serving as an output projection layer. Because we have scalar and vector features, the projection layer should embed both of them into hidden scalar and vector features while maintaining equivariance. We discuss our specific choice of projection function in Section 3.

**Projecting into queries, keys, and values** Similar to transformers, we project tokens into queries, keys, and value triplets. Because our model includes equivariant and invariant streams, we need to obtain queries, keys, and values for invariant and equivariant features. To this end, we utilize the equivariant projection to emit scalar and vector queries $\mathbf{q}_i^{inv} \in \mathbb{R}^d$, $\mathbf{q}_i^{eqv} \in \mathbb{R}^3$ as $\mathbf{q}_i^{inv}, \mathbf{q}_i^{eqv} = \phi_Q(\mathbf{x}_i, \mathbf{f}_i)$ for $i$-th token, and similarly key and value projections $\phi_K, \phi_V$ for scalar and vector keys $\mathbf{k}_i^{inv}, \mathbf{k}_i^{eqv}$ and values $\mathbf{v}_i^{inv}, \mathbf{v}_i^{eqv}$ respectively.

### 2.2. Geometric long convolution

Geometric long convolution serves as the global context aggregation module in the Geometric Hyena, analogous to the self-attention mechanism in transformer architectures. Its primary function is to encode global relations between queries and keys. The geometric long convolution comprises two components: scalar long convolution that accommodates global relations between scalar invariant tokens and equivariant vector long convolution that handles global relations between vector tokens. The scalar and vector convolutions can be used separately for global context aggregation or they can be combined into the geometric long convolution enabling the interaction between invariant and equivariant subspaces.

**Scalar long convolution** For global context aggregation of invariant scalar features, we employ long convolution (Romero et al., 2021; Poli et al., 2023) between query and key tokens. Queries serve as the input signal projection, while keys form a data-controlled implicit filter. To reduce computational complexity, the circular FFT-convolution is used. Let $\mathbf{q}^{inv}$ and $\mathbf{k}^{inv}$ be two signals of length $N$ composed of sequences of one-dimensional invariant queries $(q_i^{inv})_{i=1}^N$ and keys $(k_i^{inv})_{i=1}^N$ respectively. Then, the global context for scalar tokens is aggregated by the FFT-convolution as:

$$\mathbf{u}^{inv} = \mathbf{q}^{inv} \circledast \mathbf{k}^{inv} = \mathbf{F}^H \text{diag}(\mathbf{F}\mathbf{k}^{inv})\mathbf{F}\mathbf{q}^{inv} \quad (2)$$

where $\mathbf{F}$ is a discrete Fourier transform matrix, and $\text{diag}(\mathbf{F}\mathbf{k}^{inv})$ is a diagonal matrix containing Fourier transform of the implicit filter $\mathbf{k}^{inv}$. In the case when query's and key's dimension $d > 1$, the scalar FFT-convolution runs separately for each dimension, rendering computational complexity of $O(dN \log N)$ sub-quadratic in sequence length. This scalar long convolution is the core mechanism of the Hyena architecture (Poli et al., 2023).

**Vector long convolution** To aggregate global context for geometric tokens, we introduce an *equivariant vector long convolution*. Unlike a scalar convolution that uses dot products, vector convolution operates with vector cross product, denoted as $\times$, between vector signals. For a vector signal consisting of $N$ vector tokens $\mathbf{q}^{eqv} \in \mathbb{R}^{N \times 3}$ and a vector kernel $\mathbf{k}^{eqv} \in \mathbb{R}^{N \times 3}$, we define the vector long-convolution as:

$$\mathbf{u}_i^{eqv} = (\mathbf{q}^{eqv} \circledast_\times \mathbf{k}^{eqv})_i = \sum_{j=1}^N \mathbf{q}_i^{eqv} \times \mathbf{k}_{j-i}^{eqv} \quad (3)$$

A naive implementation of convolution is Equation 3 implies quadratic complexity. To obtain an efficient FFT-based $O(N \log N)$ implementation, recall that we can rewrite the cross-product between two vectors $\mathbf{a}, \mathbf{b} \in \mathbb{R}^3$ component-wise as $(\mathbf{a} \times \mathbf{b})[l] = \varepsilon_{lhp}\mathbf{a}[h]\mathbf{b}[p]$ where $\varepsilon_{lhp}$ is *Levi-Civita* symbol with $l, h, p \in \{0, 1, 2\}$ and $\mathbf{a}[h]$ denotes $h$-th component of vector $\mathbf{a}$. Substituting this identity into Equation 3 gives a *scalar-convolution decomposition* of the vector long-convolution:

$$\left(\mathbf{q}^{eqv} \circledast_\times \mathbf{k}^{eqv}\right)_i[l] = \sum_{j=1}^N \sum_{h,p} \varepsilon_{lhp}\, q_i^{eqv}[h]\, k_{j-i}^{eqv}[p]$$
$$= \sum_{h,p} \varepsilon_{lhp} \left(q^{eqv}[h] \circledast k^{eqv}[p]\right)_i \quad (4)$$

Thus, we can obtain $l$-th output component of a resulting vector signal as a signed sum of scalar convolutions between the Cartesian components $\mathbf{q}^{eqv}[h]$ and $\mathbf{k}^{eqv}[p]$. To obtain the $l$-th output component, we only need the index pairs $(h, p)$ for which $\varepsilon_{lhp} \neq 0$. Each $l \in \{0, 1, 2\}$ has exactly two such pairs, so the entire operation reduces to six scalar convolutions in total. Since the scalar convolution can be implemented with the FFT, decomposing the vector convolution to the series of scalar convolutions reduces its computational complexity from quadratic to $O(N \log N)$.

**Invariant-Equivariant subspace interaction**  Scalar and vector long convolutions aggregate global context for scalar and vector tokens separately, limiting the geometric complexity of context relations that can be modeled. To address this limitation, we introduce geometric long convolution, which combines scalar and vector convolutions to enable scalar-vector interactions. Our approach expands possible interactions between scalar-vector tuples $(\alpha_1, \mathbf{r}_1)$ and $(\alpha_2, \mathbf{r}_2) \in \mathbb{R} \times \mathbb{R}^3$ beyond basic scalar $\alpha_1 \alpha_2$ and vector cross $\mathbf{r}_1 \times \mathbf{r}_2$ products. In addition, we incorporate scalar-vector products $\alpha_1 \mathbf{r}_2$, $\alpha_2 \mathbf{r}_1$ and vector dot products $\mathbf{r}_1^T \mathbf{r}_2$ to represent invariant-equivariant token interactions. With this, we define the mapping from input tokens to resulting scalar and vector tokens as $\alpha_3 = \lambda_1 \alpha_1 \alpha_2 + \lambda_2 \mathbf{r}_1^T \mathbf{r}_2$ and $\mathbf{r}_3 = \lambda_3 \alpha_1 \mathbf{r}_2 + \lambda_4 \alpha_1 \mathbf{r}_2 + \lambda_5 (\mathbf{r}_1 \times \mathbf{r}_2)$ where $\lambda_i$ are trainable weights to learn the contribution of each product factor. Figure 5 of Appendix illustrates this information flow between input and output tokens. Our approach is reminiscent of a convolution operation based on the Clebsch-Gordan tensor product between two [type-0, type-1] high-order tensors (Rowe & Bahri, 2000). However, we maintain tokens in Cartesian space to avoid the high computational complexity associated with spherical representations. We derive the extension to higher-order steerable tensors in Appendix A.5.

To implement geometric long convolution efficiently, we first map the $d$-dimensional invariant tokens $\mathbf{f}_i$ to scalars via a linear layer: $\tilde{f}_i = \mathbf{w}^T \mathbf{f}_i$, where $\mathbf{w} \in \mathbb{R}^d$. This produces feature tuples $(\tilde{f}_i, \mathbf{x}_i) \in \mathbb{R} \times \mathbb{R}^3$ for each input token. Note that there are 2 interactions contributing to the scalar output and 3 interactions leading to vector output. These interactions, composed of scalar and cross products, can be handled using scalar and vector long convolutions as shown earlier, preserving sub-quadratic complexity. Additional implementation details are provided in Appendix A.4.1.

### 2.3. Selective gating

Geometric Hyena incorporates a selective gating mechanism to dynamically control information flow, similar to the softmax operation in self-attention. This mechanism enables the model to emphasize relevant tokens while suppressing less informative ones. The gating is implemented through the gating function $\gamma : \mathbb{R}^3 \times \mathbb{R}^d \to [0, 1]$ that predicts

soft masking value between 0 and 1 for each token. Practically, we implement the gating employing a projection layer (Section 2.1) with sigmoid activation. The gating emits a masking value $m_i$ for $i$-th token in a sequence. The masking is applied to the output of the geometric long convolution as $m_i \mathbf{u}_i^{inv}$ and $m_i \mathbf{u}_i^{eqv}$ for scalar and vector tokens.

Finally, the resulting gated tokens are integrated with value tokens $\mathbf{v}_i^{inv}$ and $\mathbf{v}_i^{eqv}$ using the element-wise multiplication for scalar tokens and cross product for vector tokens. With this, the Geometric Hynea block (Figure 2) includes equivariant projections, geometric long convolution, and gating. We provide a detailed ablation study on the contributions of each architecture ingredient in Appendix A.3.

## 3. Refining Geometric Hyena implementation

**Equivariant projection with local and global context** Inspired by the insights from planar graph data processing (Rampášek et al., 2022) where alternating global and local context processing significantly improves convergence, we found that the same holds also for geometric graphs. For projection with context, we employ a one-layer Equivariant Graph Neural Network (EGNN) (Satorras et al., 2021). This method maintains equivariance, allows interaction between invariant and equivariant subspaces, incorporates local context and easily accommodates edge features when available. Since EGNN operates over local context only, its complexity is linear with respect to number of nodes in a graph. We further improve the EGNN by including global context tokens that help summarizing the geometric graph. This is in line with the recent developments in planar GNNs, where the approach of using virtual nodes representing global context has shown to provide better convergence (Southern et al., 2024; Hwang et al., 2022).

Given $N$ input tokens $(\mathbf{x}_i, \mathbf{f}_i)_{i=1}^N$, $G$ global context tokens $\{\mathbf{g}_i, \mathbf{h}_i\}_{i=1}^G$ where $\mathbf{g}_i \in \mathbb{R}^3$ and $\mathbf{h}_i \in \mathbb{R}^d$, the equivariant projection layer follows the message passing framework with local and global messages computed as:

$$\mathbf{m}_{ij}^{loc} = \varphi_l(\mathbf{f}_i, \mathbf{f}_j, \|\mathbf{x}_i - \mathbf{x}_j\|_2, e_{ij}) \tag{5}$$

$$\mathbf{m}_{ij}^{glob} = \varphi_g(\mathbf{f}_i, \mathbf{h}_j, \log(1 + \|\mathbf{x}_i - \mathbf{g}_j\|_2)) \tag{6}$$

where $\varphi_g, \varphi_l : \mathbb{R}^d \times \mathbb{R}^d \times \mathbb{R}^1 \times \mathbb{R}^e \to \mathbb{R}^d$ are the functions to compute global and local messages implemented as one-layer perceptrons. Then, scalar and vector features are updated as:

$$\hat{\mathbf{x}}_i = \mathbf{x}_i + \frac{1}{|\mathcal{N}_r^k(i)|} \sum_{j \in \mathcal{N}_r^k(i)} (\mathbf{x}_i - \mathbf{x}_j) \varphi_x(\mathbf{m}_{ij}^{loc}) \tag{7}$$

$$\hat{\mathbf{f}}_i = \varphi_f(\mathbf{f}_i, \mathbf{m}_i^{loc} + \mathbf{m}_i^{glob}) \tag{8}$$

where $\mathcal{N}_r^k(i)$ represents the top-k neighbors of the i-th token within radius $r$ and feature update functions $\varphi_f : \mathbb{R}^d \times \mathbb{R}^d \to \mathbb{R}^d$ and $\varphi_x : \mathbb{R}^d \to \mathbb{R}^1$ are parameterized as one-layer perceptrons. We use log-distances for global messages to enhance stability, addressing large absolute distances that can occur with global tokens. For simplicity, we compute messages only between local tokens and from global-to-local tokens, omitting local-to-global messages and updates to global tokens.

**Global context tokens** Global context tokens provide a snapshot of the overall structure of a geometric graph. We define global context tokens as a weighted average of scalar or vector tokens across all input tokens. Given input tokens $(\mathbf{x}_i, \mathbf{f}_i)_{i=1}^N$, a set of $G$ global tokens $\{\mathbf{g}_j, \mathbf{h}_j\}_{j=1}^G$ is computed as $\mathbf{g}_j = C_j^{-1} \sum_{i=1}^N \omega_{ij}\mathbf{x}_i$ for vector tokens and $\mathbf{h}_j = C_j^{-1} \sum_{i=1}^N \omega_{ij}\mathbf{f}_i$ for scalar tokens where $C_j = \sum_{i=1}^N \omega_{ij}$. The weighting factor $\omega_{ij}$ is a learned scalar representing the contribution of the $i$-th input token to the $j$-th global context token. To detach the learning process from specific sequence length requirements, we employ a small SIREN network (Sitzmann et al., 2020) to predict $\omega_{ij}$ for each position.

**Key-Value normalization** Key-Value normalization is crucial for maintaining numerical stability in Geometric Hyena. Let $M$ be the maximum magnitude of an element in the query $\mathbf{q}$, key $\mathbf{k}$, or value $\mathbf{v}$ tokens. With this, the output magnitude after the long convolution can be bounded as $\|\mathbf{u}\|_2 \leq \|\mathbf{q}\|_2\|\mathbf{k}\|_2 \leq M^2$. When further multiplied with the values, the final output magnitude is bounded as $\|\mathbf{y}\|_2 \leq \|\mathbf{u}\|_2\|\mathbf{v}\|_2 \leq M^3$. This cubic growth with respect to input magnitudes can cause numerical instability and data type overflow. Consequently, training with the naive implementation requires extremely small learning rates and gradient clipping, resulting in slow convergence. To address this, we employ key-value normalization, ensuring unit norm for keys and values: $\|\mathbf{k}\|_2 = \|\mathbf{v}\|_2 = 1$. This bounds the output magnitude as $\|\mathbf{y}\|_2 \leq \|\mathbf{q}\|_2 \leq M$ removing the unstable cubic bound. In practice, we observed using key-value normalization to be a critical step to ensure convergence and numerical stability of Geometric Hyena.

## 4. Experiments

### 4.1. Runtime and memory benchmarks

We benchmark the runtime and peak memory consumption for a forward pass of Geometric Hyena against other equivariant models with global context: Vector Neuron Transformer (Assaad et al., 2022), Equiformer (Liao & Smidt, 2023). Additionally, we compare with the G-Transformer baseline which follows the information flow of Geometric Hyena but relies on equivariant self-attention instead of long convolutions. The G-transformer baseline is detailed in Ap-

pendix A.4.1. Similar to (Dao et al., 2022; Poli et al., 2023), we use sample sequences, and we test a one-layer model for the runtime and peak memory consumption. We record all runtimes on NVIDIA A10G GPU with CUDA 12.2.

The comparison is reported in Figure 1, showing forward pass time in milliseconds and peak GPU memory consumption in gigabytes. Geometric Hyena easily scales to longer sequences whereas VNT, Equiformer, and G-Transformer are from $20\times$ to $120\times$ slower for a sequence length of $30k$ tokens. Regarding memory usage, our model requires $50\times$ less GPU memory than self-attention-based methods for sequence length of $30k$ tokens. Moreover, we observed that when Vector Neuron Transformer and G-Transformer run out of memory on $> 37k$ tokens, *our model supports up to 2.7M tokens on a single GPU allowing for $72\times$ longer geometric context*. This memory efficiency is attributed to the FFT long convolution that avoids materializing a quadratic self-attention matrix.

### 4.2. Geometric associative recall

Associative recall is a mechanistic interpretability task used to evaluate contextual learning capabilities of sequence models (Olsson et al., 2022). In this task, a model must recall and copy bigrams it has seen before. For example, if the model encounters "Harry Potter" once, it should predict "Potter" the next time it sees "Harry" (Gu & Dao, 2023). Recent studies demonstrate that strong associative recall correlates with better generalization on real-world downstream tasks (Poli et al., 2024). Building on its widespread use for standard sequence models, we extend associative recall to geometric sequences of vectors in order to evaluate the contextual learning capabilities of equivariant models.

In geometric associative recall, each key-value bigram is linked by a rotation matrix. Given a sequence of $N$ vector tokens ending with a query token, the model retrieves the associated value based on its earlier occurrence (see Figure 4). Since rotating the entire sequence alters only the orientation of the predicted vector but not the key-value relationship, the task is rotation-equivariant. Its complexity depends on the sequence length and the size of the vocabulary, where each vocabulary entry represents a unique key-value bigram.

**Implementation details** We compare standard Hyena and Transformer models, as well as global-context equivariant models—Vector Neuron Transformer (VNT) (Deng et al., 2021; Assaad et al., 2022) and Equiformer (Liao & Smidt, 2023) to our proposed Geometric Hyena. We also include a G-Transformer baseline (Appendix A.4.1), which mirrors the information flow of Geometric Hyena but uses equivariant self-attention instead of long convolutions. We evaluate each model on 3D vector token sequences of lengths ranging from $2^7$ to $2^{10}$, and we also study how different hidden di-

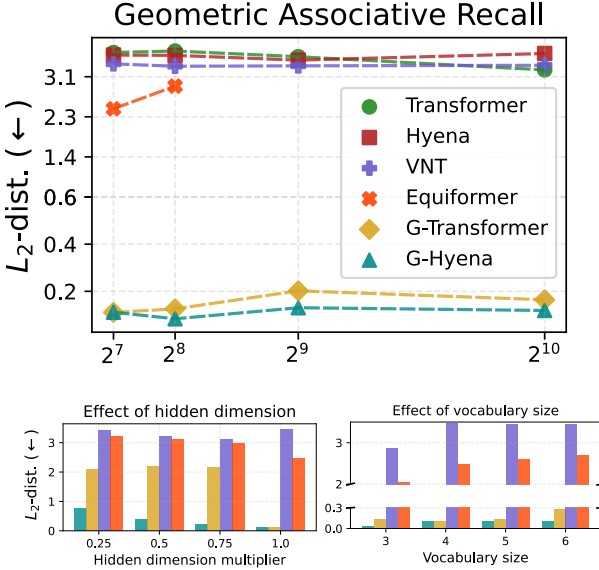

*Figure 3.* **Top:** The $L_2$-distance ($\downarrow$) between retrieved and target vectors for the geometric associative recall task over various sequence lengths. **Bottom:** The study of geometric associative recall performance of different models across varying hidden dimensions and vocabulary size.

mensions and vocabulary sizes affect the performance. We vary the hidden dimension by multiplying it by factors of $[0.25, 0.5, 0.75, 1.]$, and we vary the vocabulary size from 3 to 6. For each experiment, we sample 2600 sequences for training, and 200 sequences each for validation and testing. We use $L_2$-distance between predicted and ground truth vectors as a performance measure.

Our model uses two Geometric Hyena blocks and with a hidden dimension of 80. For the other models, we choose hyperparameters so their depth and total parameter count match those of Geometric Hyena, ensuring a fair comparison. All models are trained for 400 epochs using the Adam optimizer with a cosine learning rate scheduler and linear warm-up of 10 epochs (see Appendix A.2 for further details). We evaluate Equiformer only on sequences of lengths $2^7$ and $2^8$ as it runs out-of-memory for longer sequences.

**Results** Figure 3 (top) shows the performance on sequences of different lengths. Both Geometric Hyena and G-Transformer successfully learn to retrieve the correct value vector for a given query, while VNT and Equiformer struggle but still outperform non-equivariant baselines which merely learn the dataset mean. We attribute the strong performance of Geometric Hyena to its alternating use of local and global context. This design allows it to first form local key-value associations and then compare them across the entire sequence. This is also reflected in the strong results of G-Transformer, where we used similar building blocks

but employed equivariant self-attention instead of geometric long convolution.

Figure 3 (bottom) reports results for varying hidden dimension and vocabulary size. Geometric Hyena learns the retrieval mechanism efficiently even with a small hidden dimension, whereas other models require larger dimensions, and thus more parameters, to perform well. In particular, G-Transformer only excels at higher dimensions, and VNT and Equiformer do not perform as strongly even at larger dimensions. As vocabulary size grows, all models deteriorate, which is expected because there are more possible key-value associations and fewer occurrences of each bigram. Nevertheless, Geometric Hyena consistently outperforms the G-Transformer on all vocabulary sizes except for vocabulary size 4 where they perform on par.

### 4.3. Large molecule property prediction

We evaluate Geometric Hyena on a large-molecule property prediction task, focusing on RNA molecules represented as atomic systems with equivariance to rotations and translations. Because of RNA's rapidly growing therapeutic importance, accurate in-silico modeling can substantially reduce expensive and time-consuming in-vitro and in-vivo experiments (Prakash et al., 2024; Yazdani-Jahromi et al., 2025). We focus on RNA molecules as they pose unique challenges: they have thousands of atoms, and the lack of robust backbone extraction methods often necessitates all-atom processing (Sponer et al., 2018). Moreover, their complex secondary and tertiary structures rely on both short- and long-range interactions (Xin et al., 2008). Consequently, an effective model must provide equivariance, incorporate both local and global context, and scale to high-dimensional RNA systems.

**Datasets** We use two real-world RNA datasets used for vaccine development: Open Vaccine Covid-19 (Das et al., 2020) and Ribonanza-2k (He et al., 2024). Open Vaccine consists of 3000 RNA sequences annotated with the stability profiles (reactivity, degradation at pH10, and degradation at Mg pH10) per nucleotide which a model is trained to predict. The Open Vaccine dataset contains sequences up to 7800 nodes in all-atom resolution. Ribonanza-2k includes 2260 sequences annotated with per nucleotide reactivity profiles for DMS and 2A3 chemical modifiers (Low & Weeks, 2010). The Ribonanza-2k dataset contains RNA sequences of up to 11300 atoms. We obtain 3D all-atom structure representation using RhoFold (Shen et al., 2022), and we additionally evaluate models on eta-theta pseudotorsional backbone (Wadley et al., 2007) representation. Detailed dataset preparation is in Appendix A.2.

**Implementation details** We use atom and nucleotide identities as invariant and 3D coordinates as equivariant features.

*Table 1.* The RMSE (↓) for two large RNA molecule stability and degradation prediction tasks over the all-atom and the backbone representations. Local context methods are in red, and global context methods are in cyan. OOM denotes out-of-memory.

| Model | Open Vaccine Covid-19 (Das et al., 2020) | Ribonanza-2k (He et al., 2024) |
|---|---|---|
| *Backbone representation* | | |
| SchNet | $0.515_{\pm0.005}$ | $0.911_{\pm0.008}$ |
| TFN | $0.522_{\pm0.006}$ | $0.927_{\pm0.006}$ |
| EGNN | $0.529_{\pm0.006}$ | $0.943_{\pm0.008}$ |
| FastEGNN | $0.519_{\pm0.012}$ | $0.912_{\pm0.032}$ |
| LEFTNet | $0.502_{\pm0.004}$ | $0.889_{\pm0.008}$ |
| TMD-ET | $0.500_{\pm0.006}$ | $0.781_{\pm0.006}$ |
| Transformer | $0.400_{\pm0.004}$ | $0.637_{\pm0.006}$ |
| Hyena | $0.447_{\pm0.037}$ | $0.810_{\pm0.124}$ |
| VNT | $0.401_{\pm0.005}$ | $0.659_{\pm0.005}$ |
| G-Transformer | $0.412_{\pm0.058}$ | $0.537_{\pm0.007}$ |
| Equiformer | $0.409_{\pm0.008}$ | $0.649_{\pm0.004}$ |
| **G-Hyena** | $\mathbf{0.363}_{\pm0.045}$ | $\mathbf{0.529}_{\pm0.005}$ |
| *All-atom representation* | | |
| SchNet | $0.512_{\pm0.005}$ | $0.891_{\pm0.008}$ |
| TFN | $0.510_{\pm0.004}$ | $0.910_{\pm0.011}$ |
| EGNN | $0.511_{\pm0.005}$ | $0.928_{\pm0.022}$ |
| FastEGNN | $0.498_{\pm0.004}$ | $0.873_{\pm0.010}$ |
| LEFTNet | $0.501_{\pm0.005}$ | $0.880_{\pm0.008}$ |
| TMD-ET | $0.494_{\pm0.009}$ | $0.855_{\pm0.002}$ |
| Transformer | $0.399_{\pm0.004}$ | $0.633_{\pm0.007}$ |
| Hyena | $0.393_{\pm0.013}$ | $0.605_{\pm0.017}$ |
| VNT | $0.391_{\pm0.013}$ | $0.638_{\pm0.008}$ |
| G-Transformer | $0.391_{\pm0.071}$ | $0.592_{\pm0.043}$ |
| Equiformer | OOM | OOM |
| **G-Hyena** | $\mathbf{0.339}_{\pm0.004}$ | $\mathbf{0.546}_{\pm0.006}$ |

The models are trained to regress stability and degradation profiles per-nucleotide. The root mean squared error (RMSE) is used as evaluation metric averaged over nucleotides as in Das et al. (2020).

We compare Geometric Hyena against non-equivariant Hyena and an encoder-Transformer, both trained with data augmentation, as well as against other equivariant models with either local or global context. As local equivariant baselines, we employ SchNet (Schütt et al., 2017), TFN (Thomas et al., 2018), EGNN (Satorras et al., 2021), and recent Torch-MD Equivariant Transformer (TMD-ET) (Pelaez et al., 2024), LEFTNet (Du et al., 2023) and FastEGNN (Zhang et al., 2024). As global equivariant baselines, we employ Vector Neuron Transformer (VNT) (Assaad et al., 2022), Equiformer (Liao & Smidt, 2023) and the G-Transformer model (Appendix A.6). We use Geometric Hyena with 3 blocks and the hidden dimension of 80. For the baselines, we choose hyperparameters to equate the depth and number of parameters to the Geometric Hyena model. We train all models for 200 epochs with the Adam optimizer with

a cosine learning rate scheduler and linear warm-up. The RMSE averaged over annotated nucleotides is used as a loss function. Appendix A.2 provides further details on the setup. Additionally, in Appendix A.3 we provide a detailed ablation study on the usefulness of different components of the Geometric Hyena architecture.

**Results** Table 1 reports the RMSE averaged over nucleotides on the Open Vaccine and Ribonanza-2k datasets, using three fixed random seeds. Methods that capture global context significantly outperform those relying on local context only, highlighting the importance of modeling global geometric structure. Among the global methods, Geometric Hyena achieves the best performance on both datasets, for both all-atom and backbone representations. On Open Vaccine, it reduces RMSE by 12% and 15% for the backbone and all-atom representations, respectively, compared to the second-best model. On Ribonanza-2k, it lowers RMSE by 1.5% and 8.4% for backbone and all-atom representations. An ablation study (Appendix A.3) shows that the largest improvements come from alternating local and global context. We also hypothesize that long convolutions are a natural fit for molecules with a strong sequence prior (such as RNA and proteins) as they inherently differentiate token orderings by breaking permutation symmetry.

### 4.4. Predicting RNA molecule switching factor

We further test our model's ability to capture geometric dependencies by predicting the switching factor of RNA riboswitches (Groher et al., 2018). Specifically, we predict the dynamic range of riboswitches that measures the fold-change in gene expression between the ligand-bound and unbound states. This task requires reasoning over both short- and long-range geometric dependencies, as RNA molecules are represented as atomic systems where 3D spatial configurations significantly influence their regulatory behavior.

**Dataset** We utilize the Tc-Ribo dataset (Groher et al., 2018) that contains 355 RNA molecules (up to 3850 atoms) with experimentally measured switching factors (dynamic range). Similar to Open Vaccine and Ribonanza-2k, we utilize RhoFold to obtain all-atom 3D representation, and we additionally experiment with eta-theta pseudotorsional backbone representation.

**Implementation Details** The experimental setup for RNA switching factor closely mirrors the pipeline used for RNA degradation and stability tasks. The invariant features include atomic and nucleotide identities, while atomic coordinates are used to extract equivariant features. A model regresses the switching factor as a scalar output. And the root mean squared error (RMSE) is used as the evaluation metric. We employ the similar model and training config-

*Table 2.* The RMSE (↓) for predicting RNA ribo-switching factor measurement over the all-atom and the backbone representations.

| Model | Tc-Ribo (Groher et al., 2018) | |
|---|---|---|
| | *Backbone repr.* | *All-atom repr.* |
| SchNet | $0.737_{\pm0.002}$ | $0.691_{\pm0.018}$ |
| TFN | $0.733_{\pm0.003}$ | $0.710_{\pm0.009}$ |
| EGNN | $0.728_{\pm0.001}$ | $0.729_{\pm0.002}$ |
| FastEGNN | $0.704_{\pm0.005}$ | $0.727_{\pm0.011}$ |
| LeftNet | $0.749_{\pm0.006}$ | $0.750_{\pm0.004}$ |
| TMD-ET | $0.750_{\pm0.004}$ | $0.751_{\pm0.003}$ |
| Transformer | $0.556_{\pm0.001}$ | $0.553_{\pm0.002}$ |
| Hyena | $0.560_{\pm0.002}$ | $0.569_{\pm0.001}$ |
| G-Transformer | $0.554_{\pm0.003}$ | $0.553_{\pm0.001}$ |
| Equiformer | $\mathbf{0.550}_{\pm0.009}$ | OOM |
| **G-Hyena** | $\mathbf{0.548}_{\pm0.008}$ | $\mathbf{0.548}_{\pm0.001}$ |

*Table 3.* The MSE (↓) of predicted and ground truth all-atom and backbone protein MD trajectories.

| Model | ProteinMD (Han et al., 2022) | |
|---|---|---|
| | *Backbone repr.* | *All-atom repr.* |
| Linear | $2.27_{\pm0.001}$ | $2.90_{\pm0.001}$ |
| RF | $2.26_{\pm0.001}$ | $2.85_{\pm0.001}$ |
| TFN | $2.26_{\pm0.002}$ | $2.85_{\pm0.002}$ |
| EGNN | $2.25_{\pm0.001}$ | $2.72_{\pm0.003}$ |
| FastEGNN | $1.84_{\pm0.002}$ | NaN |
| Transformer | $75.83_{\pm6.35}$ | $79.68_{\pm24.2}$ |
| Hyena | $48.94_{\pm9.03}$ | $55.34_{\pm5.51}$ |
| G-Transformer | $2.45_{\pm0.037}$ | $3.67_{\pm0.640}$ |
| Equiformer | OOM | OOM |
| **G-Hyena** | $\mathbf{1.80}_{\pm0.009}$ | $\mathbf{2.49}_{\pm0.037}$ |

urations as in the stability and degradation experiment 4.3 except for using smaller hidden dimension of 30 and larger weight decay of 0.05.

**Results**   Table 2 reports the RMSE averaged over three random seeds for the Tc-Ribo dataset. As observed in the Open Vaccine and Ribonanza-2k datasets, methods incorporating global context consistently outperform baselines limited to local context. Among the equivariant approaches, Geometric Hyena and Equiformer achieve the best performance, slightly edging out other global equivariant methods and showing significant improvements over local ones. Interestingly, we observed that a standard non-equivariant Hyena and Transformer, when trained with data augmentation, perform competitively even with global equivariant methods, suggesting potentially lower geometric complexity of the dataset.

### 4.5. Protein molecular dynamics

Next, we evaluate Geometric Hyena on equivariant protein molecular dynamics (MD) prediction task. This task involves modeling the equilibrium dynamics of protein backbone atoms over time, where interactions between atoms are determined by both local and long-range geometric relationships.

**Dataset**   The protein molecular dynamics dataset is processed from MDAnalysis (Han et al., 2022), derived from an AdK equilibrium MD trajectory (Seyler & Beckstein, 2017). The dataset consists of 4186 proteins tertiary structures with corresponding trajectories. A time span of the MD is set to $\Delta t = 15$ as in Zhang et al. (2024). Experiments are conducted on both backbone and all-atom versions of the dataset, where the backbone version contains 855 atoms per molecule, and the all-atom version comprises 3341 atoms.

**Implementation Details**   A model predicts the spatial displacement of backbone atoms in the next MD frame as a 3D vector for each atom. The MSE between the predicted and ground-truth displacements is used as both the loss function and evaluation metric. Our model consists of 2 Geometric Hyena blocks with a hidden dimension of 32 for backbone version, and 3 Geometric Hyena blocks with a hidden dimension of 50 for all-atom version. Baseline methods are configured to have comparable parameter counts and depth. The optimization parameters are set as in Experiment 4.3. We do not run VNT, LeftNet, and TMD-ET models as their publicly available implementations do not permit vector-valued prediction and hence are not applicable for this task.

**Results**   Table 3 reports the MSE, averaged over three fixed random seeds, on the protein MD dataset. For MD trajectory prediction, Geometric Hyena reduces the final MSE by 2% for backbone and by 9% for all-atom representation compared to the second-best performing model. For FastEGNN, we observed severe numerical instabilities in the all-atom setting, leading to NaN outputs.

## 5. Conclusions

We introduce Geometric Hyena being, to the best of our knowledge, the first equivariant long-convolutional architecture for efficient modeling of global geometric context at sub-quadratic complexity. Through experiments on the novel geometric associative recall, large RNA molecule property prediction, and full protein molecular dynamics, we demonstrate that equivariant long-convolutional models can outperform other local and global equivariant methods while requiring a fraction of the computational and memory cost of self-attention. By efficiently capturing global geometric context at scale, our method opens the door to a wide

range of future applications across biological, chemical, and physical domains.

**Limitations and Future work**  This work introduces a new approach for modeling global geometric context with equivariance, tested on geometric graphs from biological or chemical systems with a canonical ordering (Jochum & Gasteiger, 1977). Unlike self-attention, the FFT-based convolution at the core of our method is not permutation equivariant (except for cyclic shifts). This property can actually be beneficial for important areas such as biomolecular modeling (e.g.-modeling RNA or proteins), where the natural ordering provides an important inductive bias. In settings like point cloud processing, though, establishing a canonical order is challenging, and the lack of permutation equivariance can become critical. Addressing this limitation by incorporating explicit permutation equivariance or learning it from data (Moskalev et al., 2023) could extend the benefits of our approach to even broader range of applications.

## Impact Statement

This paper presents work whose goal is to advance the field of Machine Learning. There are many potential societal consequences of our work, none which we feel must be specifically highlighted here.

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

# A. Appendix

## A.1. Related work

**Equivariance** Equivariance to group transformations, particularly rotations and translations in 3D, is crucial for modeling physical systems (Zhang et al., 2023). Schütt et al. (2017) condition continuous convolutional filters on relative distances to build model invariant to rotations and translations. Thomas et al. (2018); Fuchs et al. (2020); Brandstetter et al. (2021); Liao & Smidt (2023); Bekkers et al. (2023) utilize spherical harmonics as a steerable basis which enables equivariance between higher-order representations. Since computing spherical harmonics can be expensive, Jing et al. (2021b;a); Satorras et al. (2021); Deng et al. (2021) focus on directly updating vector-valued features to maintain equivariance, while Zhdanov et al. (2024a) employ another equivariant network to implicitly parameterize steerable kernels. Another recent line of work (Ruhe et al., 2023a;b; Brehmer et al., 2023; Zhdanov et al., 2024b) employs geometric algebra representation which natively provides a flexible framework for processing symmetries in the data (Dorst et al., 2009).

While these works focus on how to build equivariance into a neural network, in this paper we focus on efficient equivariance to model global geometric contexts at scale.

**Modeling geometric context** Various strategies are employed to process context information in geometric data. Convolutional methods aggregate context linearly within a local neighborhood, guided by either a graph topology (Kipf & Welling, 2016; Vadgama et al., 2025) or spatial relations in geometric graphs (Schütt et al., 2017; Wu et al., 2019; Thomas et al., 2018). Message-passing framework (Gilmer et al., 2017) generalizes convolutions, facilitating the exchange of nonlinear messages between nodes with learnable message functions. These approaches are favored for their simplicity, balanced computational demands, and expressiveness (Wu et al., 2020). However, they are limited to local interactions and are known to suffer from oversmoothing (Rusch et al., 2023). This hinders building deep message-passing networks capable of encompassing a global geometric context in a receptive field, a constraint particularly critical for modeling large geometric systems (Xu et al., 2024; 2025). To address these limitations, a recent line of work focuses on architectural modifications in form of virtual nodes (Zhang et al., 2024) or using node-to-mesh message passing (Wang et al., 2024b) to enhance the capability of a model to handle long-range interactions. Being effective in a certain scenarios, these methods are constrained by their use of low-rank approximations for global interactions, limiting their expressivity when the interactions do not adhere to low-rank behavior. Another line of work adopts self-attention mechanisms for graph (Yang et al., 2021; Kreuzer et al., 2021; Kim et al., 2022; Rampášek et al., 2022) and geometric graph (Fuchs et al., 2020; Liao & Smidt, 2023; Brehmer et al., 2023) data, outperforming convolutional and message-passing approaches. Yet, the quadratic computational cost of self-attention poses significant challenges when modeling large-scale physical systems. Concurrent work by Zhdanov et al. (2025) employs ball-tree approximations for global interactions to design equivariant self-attention, mitigating the computational overhead of large-scale systems. However, this approach may constrain the expressivity of the modeled interactions, particularly when capturing complex global dependencies beyond the approximation's scope.

In this work, we aim to develop a method for global geometric context processing with sub-quadratic computational complexity.

**State-space and long-convolutional models** The quadratic computational complexity of self-attention has driven the exploration of alternatives for modeling long context. Structured state-space models (Gu et al., 2021b) have emerged as a promising alternative, integrating recurrent and convolutional mechanisms within a single framework. These models enable parallelized training in a convolutional mode and maintain linear complexity with respect to sequence length in a recurrent mode. Models like S4 (Gu et al., 2021a), H3 (Fu et al., 2022), and Mamba (Gu & Dao, 2023; Li et al., 2024) have consistently matched or exceeded transformer performance in diverse tasks such as genomics (Schiff et al., 2024), long-range language (Wang et al., 2024a), and vision tasks (Zhu et al., 2024). Concurrently, another line of work integrates long-convolutional framework with implicit filters (Sitzmann et al., 2020; Romero et al., 2021; Zhdanov et al., 2024a) to capture global sequence context. The implicit filter formulation allows for data-controlled filtering similar to transformers, while FFT-based long convolution enables global context aggregation in sub-quadratic time (Poli et al., 2023). Such models have shown competitive performance comparable to state-space and transformer architectures in time-series modeling (Romero et al., 2021), genomics (Nguyen et al., 2024), and vision tasks (Poli et al., 2023).

Although state-space and long-convolutional methods dramatically reduced the computational costs associated with processing long sequences, their application to geometric data with equivariance remains unexplored. In this work, we take inspiration from the recently proposed Hyena operator (Poli et al., 2023) to incorporate SE(3) equivariance. To the best of our knowledge, this is the first equivariant long-convolutional model that processes global geometric contexts with sub-quadratic memory and time requirements.

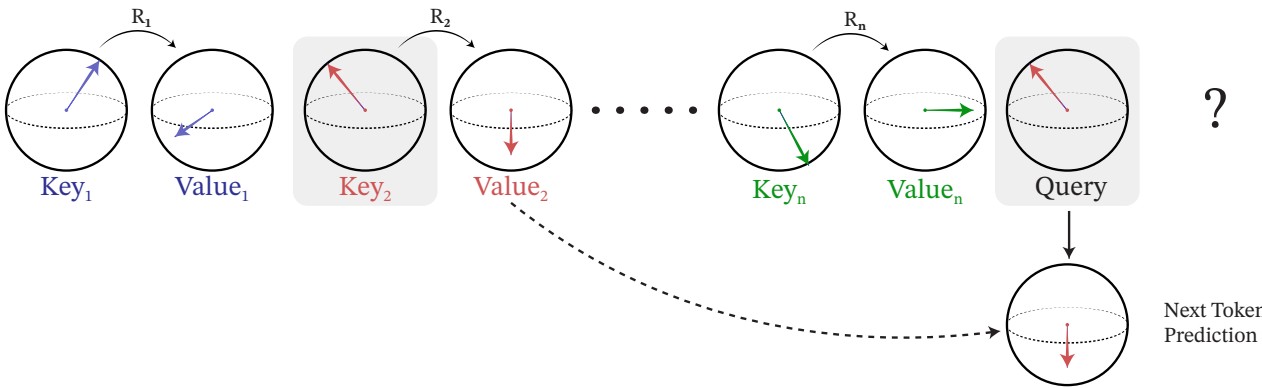

*Figure 4.* **Geometric associative recall task.** A geometric sequence consists of key and value vector tokens, where consecutive key-value pairs form bigrams. Geometric associative recall requires retrieving the value vector corresponding to a query, where the query matches one of the keys in the sequence.

## A.2. Additional experimental results and details

### A.2.1. GEOMETRIC ASSOCIATIVE RECALL

**Data generation**   Each associative recall sequence is generated by sampling bigrams (key-value pairs) from the bigram vocabulary. The size of vocabulary `vocab_size` denotes a total number of unique bigrams. The key and value in a bigram relate through a rotation matrix. Following (Olsson et al., 2022), the vocabulary is generated *on a fly* for each training sequence. The vocabularies are generated as follows: one key-value bigram consists of two vectors with orientations sampled as random unit 3D vectors from an isotropic normal distribution, and with magnitudes randomly sampled from a uniform distribution in a range of $[1, \text{vocab\_size}]$. When generating sequences from a vocabulary, the last token serves as a query as illustrated in Figure 4. We additionally constraint the generation so the target key-value pair is present in a sequence at least once.

**Models**   The vector tokens are used as the input to the equivariant branch, while for the invariant branch, we use positional encoding features (Vaswani et al., 2017) of dimension 16 as the input. The Geometric Hyena model concludes with a global pooling of equivariant features into one vector prediction. With two Geometric Hyena blocks and a hidden dimension of 80, our model ends with $480k$ trainable parameters. We configure other baseline models to match the depth and and parameter count Geometric Hyena. The local neighborhood of each token in EGNN projection layer is defined as its one immediate preceding and one subsequent tokens.

**Training**   We train all models for 400 epochs with a batch size of 8. We employ Adam optimizer (Kingma & Ba, 2014) with an initial learning rate of 0.001 and cosine learning rate scheduler (Loshchilov & Hutter, 2016) with 10 epochs of linear warmup. The weight decay is set to 0.00001. Euclidean distance is used as the loss function.

### A.2.2. LARGE RNA MOLECULE PROPERTY PREDICTION

**Open Vaccine and Ribonanza-2k datasets**   The Open Vaccine COVID-19 dataset (Das et al., 2020) provides RNA nucleotide sequences with signal-to-noise ratio (SNR) and per-nucleotide stability profile annotations, and annotated secondary structure. We first filter out sequences with an SNR lower than 1. The resulting dataset contains mRNA sequences up to 130 nucleotides which translates up to 7800 atoms. The Ribonanza dataset (He et al., 2024) contains annotated per-nucleotide degradation profiles of approximately $168k$ RNAs across 17 sub-datasets. From these, we select the sub-dataset with the label distribution closest to a normal distribution, referred to as Ribonanza-2k. The number of atoms for an individual RNA sequence goes up to 11300. Analogous to the Open Vaccine dataset Ribonanza-2k comes with annotated secondary structures. The Tc-Ribo dataset (Groher et al., 2018) consists of 355 mRNA sequences up to 73 nucleotides and 3850 atoms, and a one regression label per molecule. For all datasets, we employ the state-of-the-art 3D structure prediction tool RhoFold (Shen et al., 2022) to get tertiary structures for RNA molecules. Compared to other existing 3D structure prediction tools which can take hours to run for even single sequences, RhoFold is reported to be more accurate and fast to run and has been used in prior works as the tool of choice (He et al., 2024).

*Table 4.* EGNN with sequence prior on Open Vaccine dataset

| Model | Open Vaccine | |
| --- | --- | --- |
| | *Backbone repr.* | *All-atom repr.* |
| Seq-EGNN | $0.527_{\pm 0.006}$ | $0.506_{\pm 0.009}$ |
| EGNN-Seq | $0.489_{\pm 0.002}$ | $0.490_{\pm 0.004}$ |

*Table 5.* Ablation study results. Checkmarks indicate the presence of a component. Lower RMSE values indicate better performance. The best result is highlighted in bold.

| Component | | | | | RMSE *(all-atom)* | |
| --- | --- | --- | --- | --- | --- | --- |
| Local context | Global context | Gating | KV norm | Geometric convolution | Open Vaccine Covid-19 | Ribonanza-2k |
| ✓ | ✓ | QK | ✓ | ✓ | **0.339** | **0.544** |
| ✓ | ✓ | K | ✓ | ✓ | 0.345 | 0.547 |
| ✓ | ✓ | | ✓ | ✓ | 0.449 | 0.551 |
| | ✓ | QK | ✓ | ✓ | 0.450 | 0.614 |
| ✓ | | QK | ✓ | ✓ | 0.421 | 0.545 |
| ✓ | ✓ | QK | | ✓ | NaN | NaN |
| ✓ | ✓ | QK | ✓ | | 0.349 | 0.557 |

Each RNA molecule is represented as a geometric graph where nodes correspond to individual atoms, and edges are defined by the point cloud representation, capturing the spatial arrangement of these atoms. The node features have a dimension of 14, encoding the atomic number as an integer and including a one-hot encoding for the elements (H,C,N,O,P), and we also use nucleotide identities and nucleotide loop type, estimated by RhoFold, as input features.

**Models and training**   When predicting per-nucleotide properties from atom-level representation, we employ nucleotide sum pooling which sums atom features within each nucleotide in the last layer before making a prediction. For global readout, we directly use sum pooling on atom features. For the Open Vaccine and Ribonanza-2k datasets, we follow (Wayment-Steele et al., 2022) and train all models in a multi-task manner, predicting all properties simultaneously using average root mean squared error (RMSE) as the loss function. Training runs for 200 epochs with a batch size of 16 for Open Vaccine and Tc-Ribo and 10 for Ribonanza-2k. We employ the Adam optimizer (Kingma & Ba, 2014) with an initial learning rate of 0.001, using a cosine learning rate schedule with a 10-epoch of linear warm-up. Weight decay is set to 0.0005 for Open Vaccine and Ribonanza-2k datasets, and to 0.05 for Tc-ribo.

### A.2.3. EXTENDING LOCAL CONTEXT WITH SEQUENCE-STRUCTURED PRIORS

Given the sequence-like structure of RNA molecules in the property prediction task, we further explore incorporating sequence-structured priors to enhance local methods. While this direction presents significant challenges and extends beyond the scope of this paper, we conducted a preliminary experiment to inspire future research. Specifically, we used a Hyena layer to extract sequence features followed by EGNN (Seq-EGNN), and also tested the reverse setup, where EGNN first extracts geometric features and a sequence model processes them afterward (EGNN-Seq). The experiment is conducted in Open Vaccine dataset.

The results, presented in Table 4, indicate that incorporating sequence priors has minimal impact when EGNN is applied to sequence features. However, applying the sequence model to EGNN-derived features shows a slight improvement, highlighting the potential for further exploration in this direction.

### A.3. Ablation study

We provide the ablation study on individual architectural components of Geometric Hyena in Table 5. We test the impact of various modules for the model used in RNA property prediction Experiment 4.3. We follow the same experimental setup

with the same models as in the main experiment but we train the models on one common random seed to reduce training and evaluation time.

As can be seen from Table 5, the optimal configuration for Geometric Hyena includes using both local and global context with the gating applied on top of the features resulting from geometric long convolution (QK). Note that using geometric long convolution with scalar-vector interaction improves the performance of the model compared to using scalar and vector long convolutions separately Also, we observe that using key-value normalization (KV-norm) is critical to ensure stable convergence of the model.

### A.4. Additional details on architectural components of Geometric Hyena

#### A.4.1. GEOMETRIC LONG CONVOLUTION BY SCALAR AND VECTOR LONG CONVOLUTIONS

Here we provide details on how the geometric long convolution can be implemented using scalar and vector long convolutions. Recall that the geometric long convolution combines scalar and vector convolutions to enable scalar-vector interactions. Given input scalar-vector tuples $(\alpha_1, \mathbf{r}_1)$ and $(\alpha_2, \mathbf{r}_2) \in \mathbb{R} \times \mathbb{R}^3$, the information flow from inputs to the output tuple $(\alpha_3, \mathbf{r}_3)$ can be schematically represented as in Figure 5. In particular, the elements of the output tuple are computed as:

$$\alpha_3 = \lambda_1 \alpha_1 \alpha_2 + \lambda_2 \mathbf{r}_1^T \mathbf{r}_2$$
$$\mathbf{r}_3 = \lambda_3 \alpha_1 \mathbf{r}_2 + \lambda_4 \alpha_2 \mathbf{r}_1 + \lambda_5 (\mathbf{r}_1 \times \mathbf{r}_2)$$

To implement this in a convolutional manner, we decompose the geometric long convolution into a series of scalar and vector long convolutions. For clarity, we omit the token index $i$ in the following equations, treating $\alpha$ and $\mathbf{r}$ as scalar and vector-valued signals of $N$ tokens, respectively.

**Scalar output computation** The scalar output $\alpha_3$ involves two terms: (i) scalar-scalar product: $\lambda_1 \alpha_1 \alpha_2$, and (ii) vector dot product: $\lambda_2 \mathbf{r}_1^T r2$. The scalar-scalar term can be directly computed using the scalar long convolution as defined in Equation 2: $\alpha_1 \circledast \alpha_2 = \mathbf{F}^H \text{diag}(\mathbf{F}\alpha_2) \mathbf{F}\alpha_1$. The vector dot product can be decomposed into three scalar convolutions, one for each dimension: $(\mathbf{r}_1^T \mathbf{r}_2)_i = \sum_{d=1}^{3} (r_1[d] \circledast r_2[d])i$ where $r_1[d]$ and $r_2[d]$ represent the $d$-th component of vectors $\mathbf{r}_1$ and $\mathbf{r}_2$, respectively. With this, the scalar part output of the geometric long convolution can be written as:

$$\alpha_3 = \lambda_1 (\alpha_1 \circledast \alpha_2) + \lambda_2 \sum_{d=1}^{3} (r_1[d] \circledast r_2[d]) \tag{9}$$

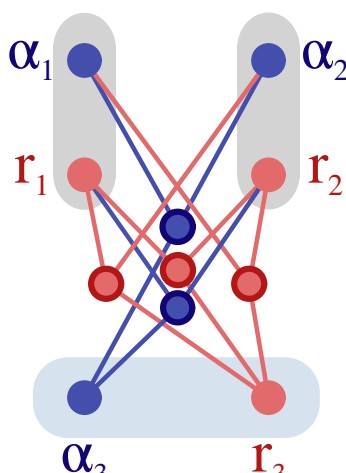

*Figure 5.* Scalar-vector interactions in geometric long convolution. **Blue** lines represent interactions leading to a scalar output $\alpha_3$, and **red** lines are interactions leading to a vector output $\mathbf{r}_3$.

**Vector output computation** The vector output $\mathbf{r}_3$ involves three terms: (i) scalar-vector product: $\lambda_3 \alpha_1 \mathbf{r}_2$, (ii) scalar-vector product: $\lambda_4 \alpha_2 \mathbf{r}_1$, and (iii) vector cross product: $\lambda_5 (\mathbf{r}_1 \times \mathbf{r}_2)$. Both scalar-vector product terms can be computed using scalar convolutions for each vector component as $\alpha_1 \circledast r_2[d]$ and $\alpha_2 \circledast r_1[d]$ for $d$-th component of the resulting vector. The vector cross product part can be computed using the vector long convolution as defined in Equation 3 of the main text. With this, the vector part output of the geometric long convolution can be written as:

$$\mathbf{r}_3 = \lambda_3 \bigsqcup_{d=1}^{3} (\alpha_1 \circledast \mathbf{r}_2[d]) + \lambda_4 \bigsqcup_{d=1}^{3} (\alpha_2 \circledast \mathbf{r}_1[d]) + \lambda_5 (\mathbf{r}_1 \circledast_\times \mathbf{r}_2) \tag{10}$$

where $\bigsqcup_{d=1}^{3}$ denotes concatenation along the $d$-th component.

A.4.2. PROOF OF EQUIVARIANCE

**Projection function**   Our framework permits any equivariant network as a projection function, making equivariance of the projection function a prerequisite. We use a modified EGNN (Satorras et al., 2021) with global messages $\mathbf{m}_{ij}^{glob} = \varphi_g\big(\mathbf{f}_i, \mathbf{h}_j, \log(1 + \|\mathbf{x}_i - \mathbf{g}_j\|_2)\big)$. This maintains E(n)-equivariance if global context tokens $\mathbf{g}_j$ are equivariant. The proof follows EGNN's, as $\mathbf{f}_i$ and $\mathbf{h}_j$ are invariant, and $\|(\mathbf{R}\mathbf{x}_i + t) - (\mathbf{R}\mathbf{g}_j + t)\|_2 = \|\mathbf{R}(\mathbf{x}_i - \mathbf{g}_j)\|_2 = \|\mathbf{x}_i - \mathbf{g}_j\|_2$.

**Global context tokens**   We next proof the equivariance of global context token mechanism. Recall that given input tokens $\{\mathbf{x}_i, \mathbf{f}_i\}_{i=1}^N$, a set of $G$ geometric tokens $\{\mathbf{g}_j, \mathbf{h}_j\}_{j=1}^G$ is obtained as $\mathbf{g}_j = C_j^{-1} \sum_{i=1}^N \omega_{ij} \mathbf{x}_i$ for vector tokens and $\mathbf{h}_j = C_j^{-1} \sum_{i=1}^N \omega_{ij} \mathbf{f}_i$ for scalar tokens where $C_j = \sum_{i=1}^N \omega_{ij}$ with $\omega_{ij}$ being learnable weights. Firstly, $\mathbf{h}_j$ maintain invariance as a function quantities $\mathbf{f}_i$ and $\omega_{ij}$ which by design only depend on the ordinal position of the token in a sequence. For the equivariance part:

$$C_j^{-1} \sum_{i=1}^N \omega_{ij}(\mathbf{R}\mathbf{x}_i + t) = C_j^{-1} \sum_{i=1}^N \omega_{ij} \mathbf{R}\mathbf{x}_i + C_j^{-1} \sum_{i=1}^N \omega_{ij} t$$
$$= \mathbf{R}(C_j^{-1} \sum_{i=1}^N \omega_{ij}\mathbf{x}_i) + \big(\sum_{i=1}^N \omega_{ij}\big)^{-1}\big(\sum_{i=1}^N \omega_{ij}\big)t$$
$$= \mathbf{R}\mathbf{g}_j + t$$

**Vector long convolution**   Given a vector signal consisting of $N$ vector tokens $\mathbf{q}^{eqv} \in \mathbb{R}^{N \times 3}$ and a vector kernel $\mathbf{k}^{eqv} \in \mathbb{R}^{N \times 3}$, recall that $i$-th element of the result if the vector long-convolution is defined as $\sum_{j=1}^N \mathbf{q}_i^{eqv} \times \mathbf{k}_{j-i}^{eqv}$. Since a cross product is already equivariant to rotations, the whole vector convolution is also equivariant to rotations provided the queries $\mathbf{q}^{eqv}$ and the keys $\mathbf{k}^{eqv}$ are rotated accordingly (which is guaranteed when the projection function is equivariant). Formally:

$$\sum_{j=1}^N \mathbf{R}\mathbf{q}_i^{eqv} \times \mathbf{R}\mathbf{k}_{j-i}^{eqv} = \sum_{j=1}^N \mathbf{R}(\mathbf{q}_i^{eqv} \times \mathbf{k}_{j-i}^{eqv}) = \mathbf{R}\sum_{j=1}^N \mathbf{q}_i^{eqv} \times \mathbf{k}_{j-i}^{eqv}$$

This provides equivariance to the SO(3). For the SE(3) equivariance, we can first center the input tokens relative to their center of mass, apply the convolution, then uncenter the result.

**Geometric long convolution**   The geometric long convolution combines scalar and vector convolutions for scalar-vector interactions. We prove its equivariance by examining each term under transformations. For the scalar output: (i) $\alpha_1 \circledast \alpha_2$ is invariant as it involves only scalar quantities; (ii) $\sum_{d=1}^3 (r_1[d] \circledast r_2[d])$ is rotation-invariant as it reassembles a dot product which is invariant to rotations. For the vector output: (i) Terms $\bigsqcup_{d=1}^3 (\alpha_1 \circledast r_2[d])$ and $\bigsqcup_{d=1}^3 (\alpha_2 \circledast r_1[d])$ transform equivariantly under rotation $\mathbf{R}$ because $\mathbf{R}(\alpha \circledast \mathbf{r}) = \alpha \circledast (\mathbf{R}\mathbf{r})$ for any scalar $\alpha$ and vector $\mathbf{r}$, as convolution distributes over vector components; (ii) $\mathbf{r}_1 \circledast_\times \mathbf{r}_2$ is equivariant as previously proven for vector long convolution. This establishes SO(3) equivariance. For SE(3) equivariance, we center input tokens to their center of mass before convolution and uncenter afterwards, making the operation translation-equivariant.

**Selective gating**   The gating function $g : \mathbb{R}^3 \times \mathbb{R}^d \to [0, 1]$ is composed of an equivariant projection layer, a linear layer, and a sigmoid activation. Since the projection layer is equivariant (as proven earlier), and both the linear layer and sigmoid operate only on top of invariant features, the gating function itself is invariant to SE(3) transformations. For scalar tokens, the gating operation $m_i \mathbf{u}_i^{inv}$ preserves invariance as it's a product of invariant terms. For vector tokens, under rotation $\mathbf{R}$, we have $m_i(\mathbf{R}\mathbf{u}_i^{eqv}) = \mathbf{R}(m_i\mathbf{u}_i^{eqv})$ since $m_i$ is a scalar. For translations, centering and uncentering steps ensure translation equivariance, obtaining the SE(3)-equivariance.

**Key-Value normalization**   We prove that the key-value normalization preserves SO(3) equivariance but not SE(3) equivariance. We normalize keys and values to unit norm: $\|\mathbf{k}\|_2 = \|\mathbf{v}\|_2 = 1$. For scalar keys and values, this normalization

is invariant to rotations as it operates on top of invariant quantities. For vector keys and values, under rotation $\mathbf{R}$, we have $\|\mathbf{R}\mathbf{k}\|_2 = \|\mathbf{k}\|_2 = 1$ and $\|\mathbf{R}\mathbf{v}\|_2 = \|\mathbf{v}\|_2 = 1$. The normalized vectors transform as $\mathbf{k}/\|\mathbf{k}\|_2 \to \mathbf{R}\mathbf{k}/\|\mathbf{R}\mathbf{k}\|_2 = \mathbf{R}(\mathbf{k}/\|\mathbf{k}\|_2)$, and similarly for $\mathbf{v}$. Thus, the normalization commutes with rotations, providing the SO(3) equivariance. For translations, centering and uncentering steps ensure translation equivariance, obtaining the SE(3)-equivariance.

### A.5. Extending Geometric Hyena to higher-order representation

To further encourage the research towards equivariant long-convolutional models for geometric systems, we outline a blueprint of the potential extension of Geometric Hyena to accommodate higher-order representations. We consider two possibilities: (i) employing spherical harmonics-based steerable representation with the tensor product, and (ii) interaction with higher-order features by scalarization trick (Satorras et al., 2021; Cen et al., 2024).

**Steerable long convolution with higher-order representations**  Following the steps used in this work to derive the cross product-based vector long convolution (Section 2.2), one can employ steerable representations and tensor products to derive a long convolution for higher-order steerable vectors. Instead of vectors in $\mathbb{R}^3$ and the cross product, we use steerable vectors and the tensor product. The long convolution can then be constructed using the tensor product and reduced to a series of scalar convolutions by factoring out the Clebsch-Gordan (CG) coefficients.

Let $\mathbf{q}^{(l_1)}$ and $\mathbf{k}^{(l_2)}$ be sequences of steerable features of degrees $l_1$ and $l_2$ respectively, each with length $N$. These features transform according to the spherical harmonics representations of SO(3). We can define steerable long convolution as:

$$\left(\mathbf{q}^{l_1} \circledast_{\mathrm{CG}} \mathbf{k}^{l_2}\right)_{i,m}^{(l)} = \sum_{j=1}^{N} \sum_{m_1=-l_1}^{l_1} \sum_{m_2=-l_2}^{l_2} C_{(l_1 m_1)(l_2 m_2)}^{(lm)} \mathbf{q}_{i,m_1}^{(l_1)} \mathbf{k}_{j-i,m_2}^{(l_2)} \tag{11}$$

where $C_{(l_1 m_1)(l_2 m_2)}^{(lm)}$ denotes Clebsch-Gordan coefficients from type $l_1, l_2$ to $l$.

To efficiently compute this convolution, similarly to geometric long convolution, we can decompose it into scalar convolutions factoring out the Clebsch-Gordan coefficients, reducing the computational complexity from to $O(N \log N)$:

$$\mathbf{u}_{i,m}^{(l)} = \sum_{m_1=-l_1}^{l_1} \sum_{m_2=-l_2}^{l_2} C_{(l_1 m_1)(l_2 m_2)}^{(lm)} \left(\mathbf{q}_{m_1}^{(l_1)} \circledast \mathbf{k}_{m_2}^{(l_2)}\right)_i \tag{12}$$

*This is a direct extension of our method*, since the cross product used in our vector long convolution corresponds to the case where $l = l_1 = l_2 = 1$, and the Clebsch-Gordan coefficients reduce to the Levi-Civita symbol.

The advantage of the fully steerable approach is that it allows interactions between features of arbitrary degrees, capturing complex geometric relationships and maintaining strict equivariance under rotations. However, this approach might be computationally expensive due to the large number of long convolutions and Clebsch-Gordan coefficients involved, especially for higher degree representation. Potentially, this computational overhead hinders the usage of fully steerable methods for large-scale geometric systems, where computational efficiency is crucial. In this paper, we purposefully chose to focus on a simpler case with cross product due to its computational efficiency. By restricting to type-1 representations, we reduce the computational complexity and make the method more practical for large-scale applications, while still capturing essential geometric properties through equivariant operations.

**Faster higher-order representations by scalarization**  Alternatively, to efficiently model interactions between high and low order features without the heavy computational cost of fully steerable methods, we can employ the scalarization trick. This approach extends the geometric long convolution framework by introducing higher-order components into the feature representations while simplifying computations by selectively considering interactions.

In this method, each feature tuple is expanded to include a higher-order part. Specifically, using notation from Section 2.2, we extend feature representation from $(\alpha, \mathbf{r})$ to $(\alpha, \mathbf{r}, \mathbf{v})$ containing scale, vector, and higher-order representation $\mathbf{v}$ respectively. When performing the convolution between two such feature tuples, we drop the higher-order to higher-order interaction terms. With this, we can focus on interactions that involve higher-order to low-order order and low-order to low-order interactions only, effectively scalarizing the processing of higher-order features.

By excluding higher-order to higher-order interactions, we significantly reduce computational complexity. The higher-order features still contribute to the model through their interactions with scalar parts, allowing the network to capture important geometric information without the full computational overhead associated with processing all higher-order interactions.

## A.6. Details on the G-Transformer baseline

Since Geometric Hyena utilizes long convolutions based on cross products, we design simple equivariant vector self-attention mechanism to align the global geometric context aggregation for Hyena and transformer models for more direct comparison. To align the information flow, we use a similar architecture to Geometric Hyena (i.e. input/output projections, QKV-projection as in Figure 2) but instead of long convolutions, we employ cross product-based equivariant self-attention to aggregate global context. Thus, the only difference between Geometric Hyena and the SE(3)-Transformer baseline is the global context aggregation mechanism.

**Equivariant dot product vector self-attention** Consider sequences of $N$ vector query, key, and value tokens denoted as $\mathbf{q}, \mathbf{k}, \mathbf{v} \in \mathbb{R}^{N \times 3}$. We first construct the dot product matrix $\mathbf{C} \in \mathbb{R}^{N \times N}$ with $\mathbf{C}_{ij} = \mathbf{q}_i^T \mathbf{k}_j$. The entries of the dot product matrix are invariant to rotations of queries and keys since $\mathbf{q}_i^T \mathbf{k}_j = \mathbf{q}_i^T \mathbf{R}^T \mathbf{R} \mathbf{k}_j$. Then, softmax is applied row-wise to compute the self-attention matrix $\mathbf{S} = \texttt{softmax}(\frac{1}{\sqrt{N}}\mathbf{C})$. The self-attention matrix is used to aggregate vector values as $\mathbf{u}_i = \sum_{j=1}^N \mathbf{S}_{ij} \mathbf{v}_j$. The rotation equivariance is preserved as $\mathbf{S}_{ij}$ is invariant scalar so $\sum_{j=1}^N \mathbf{S}_{ij} \mathbf{R} \mathbf{v}_j = \mathbf{R} \sum_{j=1}^N \mathbf{S}_{ij} \mathbf{v}_j = \mathbf{R} \mathbf{u}_i$. Equivariance to translations can be achieved by initially centering the data (subtracting the center of mass) and then re-centering the resulting tokens.

**Equivariant cross product vector self-attention** We also experiment with cross product based equivariant self-attention. We start by constructing a query-key cross product tensor $\mathbf{C} \in \mathbb{R}^{N \times N \times 3}$ where each element $\mathbf{C}_{ij} = \mathbf{q}_i \times \mathbf{k}_j$, or using Levi-Civita notation as in Eq. 4, $\mathbf{C}_{ij}[l] = \varepsilon_{lhp} \mathbf{q}_i[h] \mathbf{k}_j[p]$. To integrate a softmax selection mechanism, we first compute a matrix $\eta(\mathbf{C}) \in \mathbb{R}^{N \times N}$ containing the $L_2$ norms of all cross products, specifically $\eta(\mathbf{C})_{ij} = \|\mathbf{q}_i \times \mathbf{k}_j\|_2$. Applying softmax to $\eta(\mathbf{C})$ then determines the vector pairs to select from the cross product tensor. Lastly, the values $\mathbf{v}$ are cross-multiplied with the softmax-filtered cross product tensor. Overall, the equivariant vector self-attention reads as:

$$\mathbf{S} = \texttt{softmax}(\frac{1}{\sqrt{N}}\eta(\mathbf{C})) \odot \mathbf{C} \tag{13}$$

$$\mathbf{u}_i = \frac{1}{N} \sum_{j=1}^N \mathbf{S}_{ij} \times \mathbf{v}_j \tag{14}$$

where the softmax is applied row-wise, and $\odot$ stands for element-wise product. Consequently, $\mathbf{S} \in \mathbb{R}^{N \times N \times 3}$ represents a tensor that encapsulates a soft selection of cross products between $\mathbf{q}_i$ and $\mathbf{k}_j$. Since the tensor $\mathbf{C}$ is constructed using cross products, it naturally maintains equivariance to rotations of queries and keys. Furthermore, the softmax is applied to the $L_2$ norms of the cross products making it rotation-invariant. Consequently, the self-attention tensor $\mathbf{S}$ is a product of rotation-invariant scalar and rotation-equivariant vector quantities, rendering it rotation-equivariant. The Eq. 14 further preserves rotation-equivariance due to the inherent equivariance of the cross product. Equivariance to translations can be achieved by initially centering the data and then re-centering the resulting tokens.

Early experiments revealed that the equivariant dot product vector self-attention either performs on par or better than the cross product version while being computationally faster and less memory intensive. We hence proceeded with the equivariant dot product self-attention as a global context module for G-Transformer.

### A.6.1. PYTORCH IMPLEMENTATION

Equivariant vector long convolution is an essential building block in Geometric Hyena. It can be used as is to aggregate global geometric context across vector tokens or can be combined with standard scalar long convolution into geometric long convolution. Here, we provide a simple Pytorch implementation for the rotation-equivariant (without centering) vector long convolution in Code 1.

```
1
2    class VectorLongConv(nn.Module):
3        def __init__(self):
4            super(VectorLongConv, self).__init__()
5
6            # L cross-prod tensor factorized:
7            expand_vec_mat = torch.FloatTensor([[1, 0, 0],
8                                                [1, 0, 0],
9                                                [0, 1, 0],
10                                               [0, 1, 0],
11                                               [0, 0, 1],
12                                               [0, 0, 1]]).view(1,1,1,6,3)
13           self.register_buffer("expand_vec_mat", expand_vec_mat, persistent=False)
14
15           # H cross-prod tensor factorized:
16           cross_expand_vec_mat = torch.FloatTensor([[0 , 1, 0],
17                                                      [0 , 0,-1],
18                                                      [-1, 0, 0],
19                                                      [0 , 0, 1],
20                                                      [1 , 0, 0],
21                                                      [0 ,-1, 0]]).view(1,1,1,6,3)
22           self.register_buffer("cross_expand_vec_mat", cross_expand_vec_mat, persistent=False)
23
24           # P cross-prod tensor factorized:
25           cross_sum_mat = torch.FloatTensor([[0,0,0,1,0,1],
26                                              [0,1,0,0,1,0],
27                                              [1,0,1,0,0,0]]).view(1,1,1,3,6)
28           self.register_buffer("cross_sum_mat", cross_sum_mat, persistent=False)
29
30       def conv_fft(self, x, k):
31           N = x.shape[-2]
32           fft_x = torch.fft.rfft(x, n=N, dim=-2)
33           fft_k = torch.fft.rfft(k, n=N, dim=-2)
34           conv_xk = torch.fft.irfft(fft_x*fft_k, norm="backward", n=N, dim=-2)
35           return conv_xk
36
37       def forward(self, x, k):
38
39           # batch, channel, sequence length, 3
40           B,C,N,_ = x.shape
41
42           # expand aux matrices
43           expand_vec_mat = self.expand_vec_mat.expand(B,C,N,6,3)
44           cross_expand_vec_mat = self.cross_expand_vec_mat.expand(B,C,N,6,3)
45           cross_sum_mat = self.cross_sum_mat.expand(B,C,N,3,6)
46
47           # expand inputs and matmul
48           expanded_x = torch.matmul(expand_vec_mat, x.unsqueeze(-1)).squeeze(-1)
49           cross_expanded_k = torch.matmul(cross_expand_vec_mat, k.unsqueeze(-1)).squeeze(-1)
50
51           # fft conv
52           fft_conv_xk = self.conv_fft(expanded_x, cross_expanded_k)
53           reduced_fft_conv_cd = torch.matmul(cross_sum_mat, fft_conv_xk.unsqueeze(-1)).squeeze(-1)  / N
54
55           return reduced_fft_conv_cd
```

*Code 1.* Pytorch implementation of the equivariant vector long convolution.

