# OpenReview forum: "Geometric Hyena Networks for Large-scale Equivariant Learning"
_ICML.cc/2025/Conference — ICML 2025 spotlightposter_

### Official Review · Reviewer_XkYv · 2025-02-18

**Overall Recommendation:** 2

**Summary:**

This paper introduces an SE(3)-equivariant extension of the Hyena model (Poli et al. 2023) which employs long convolutions evaluated in the Fourier domain. This Geometric Hyena model is used for property predictions of large biological molecules. The authors show that their model outperforms transformer models in terms of runtime and memory as well as performance in several numerical experiments.

**Claims And Evidence:**

The claimed runtime- and memory benefits over transformer models are convincing. The claimed performance benefits are largely convincing. However, as the authors point out themselves, the performance benefit in the RNA property prediction task and on the geometric associative recall task is likely due to the alternating local and global context and not the long convolutions (the error bars for G-Transformer and G-Hyena overlap).

**Essential References Not Discussed:**

None.

**Experimental Designs Or Analyses:**

The experimental designs seem appropriate.

**Methods And Evaluation Criteria:**

The chosen problems largely seem to be appropriate to evaluate the model. However, I am missing a baseline of a non-equivariant state space model (e.g. the non-equivariant Hyena model), since the claimed benefit of the proposed model is the combination of the efficiency state-space models with the performance of equivariant models. In light of recent doubts about the importance of equivariance (e.g. [arXiv.2311.17932], [arXiv: 2410.24169], AlphFold3) it would be interesting to see how the Geometric Hyena compares to efficient non-equivariant models.

**Other Comments Or Suggestions:**

For some typos, see above.

**Other Strengths And Weaknesses:**

Strengths:
Strong experimental results and thorough ablations.

Weaknesses:
The presentation of the paper is seriously lacking. There are numerous typos and grammatical errors in the manuscript (e.g. "Heyna" instead of "Hyena" in the tables). The related works section is an important part of the paper and has to be part of the main text, it does not belong in the supplementary material. For removing such an important section from the main text, the manuscript seems wordy at times, e.g. the "invariant-equivariant subspace interaction" section can be shortened considerably and the beginnings of sections 4.1 and 4.2 -> "implementation details" are almost the same. Also the equations given lack precision. For instance, the domain and co-domain of $\Psi$ given above (1) do not match the sets in (1) and the hatted quantities are not introduced. The notation $F^H$ in (2) is not introduced. Similarly, the gating function $\gamma:\mathbb{R}^3\times\mathbb{R}^d\rightarrow[0,1]$ seems to act per token as it is introduced, is that correct?

**Questions For Authors:**

No further questions.

**Relation To Broader Scientific Literature:**

Equivariant architectures are a standard tool in quantum mechanical property prediction. Since state-space models are a recent development in other domains, it is interesting to consider an equivariant version of a popular state-space model.

**Theoretical Claims:**

The equivariance of the proposed layers is obvious.

---

> ### Author Rebuttal · Authors · 2025-03-31
>
> ## Computational and performance benefits:
>
> We appreciate the reviewer acknowledging the runtime and memory benefits of our method, as computational efficiency is the main purpose of our work.
>
> Second, we believe there may be a misunderstanding regarding performance improvements. G-Hyena significantly outperforms the G-Transformer model—with non-overlapping error bars on Tc-Ribo in both the backbone and all-atom settings (-7\% error reduction in both cases). It also shows substantial gains on the Protein-MD task in both backbone and all-atom scenarios (-26\% and -36\% error reduction, respectively). Since the primary architectural difference between G-Hyena and G-Transformer is the geometric long convolution, we attribute these performance improvements to that component. In addition, the associative recall experiment (Figure 3, bottom left) shows that G-Hyena scales more favorably with respect to the hidden dimension compared to G-Transformer. This improved scaling behavior is practically relevant, as it potentially enables more efficient model size reduction and further lowers memory usage. Taken together, we consider both the computational and performance advantages of our model to be significant in the context of the conducted experiments.
>
> ## Non-equivariant state space model:
> We agree that the extended comparison with the non-equivariant Hyena model will further strengthen the experiments. In addition to Hyena in the associative recall experiments, we test a standard Hyena model trained with data augmentation on Open Vaccine, Ribonanza-2k, Tc-ribo, and Protein-MD datasets.
>
> |Results (*backbone*) |Open Vaccine|Ribonanza-2k|Tc-ribo|Protein-MD|
> |-----|------------------------|------------------------|-------------------|----------------|
> |Hyena|0.447±0.037|0.810±0.124|0.560±0.002|48.94±9.03|
> |G-Hyena|**0.363±0.045**|**0.529±0.005**|**0.517±0.025**|**1.80±0.009**|
>
>
> |Results (*all-atom*)|Open Vaccine|Ribonanza-2k|Tc-ribo|Protein-MD|
> |-----|------------------------|------------------------|-------------------|----------------|
> |Hyena|0.393±0.002|0.605±0.017|0.569±0.001|55.34±5.51|
> |G-Hyena|**0.339±0.004**|**0.546±0.006**|**0.552±0.003**|**2.49±0.037**|
>
> Results suggest that standard non-equivariant Hyena trained with extensive data augmentation still significantly lags behind properly equivariant Geometric Hyena (with G-Hyena performance gain ranging from -3\% error reduction on Tc-Ribo all-atom to -53\% error reduction Ribonanza-2k backbone). Furthermore, the non-equivariant Hyena model struggles to generalize on the Protein-MD dataset where it underperforms significantly compared to equivariant methods. The results will be included in the revised version of the paper.
>
> ## Presentation of the paper and clarifications:
> - We will refine the text of the paper to correct grammatical errors and typos as soon as openreview allows revision. We will further streamline the writing in "invariant-equivariant subspace interaction" section and remove redundancies in the beginnings of sections 4.1 and 4.2.
> - In Eq. (1), $\hat{\mathbf{x}}$ and $\hat{\mathbf{f}}$ refer to the output of Geometric Hyena.
> - Indeed, the more precise domain and co-domain formulation would be $\Psi: (\mathbb{R}^{3} \times \mathbb{R}^{d})^{N} \rightarrow (\mathbb{R}^{3} \times \mathbb{R}^{d})^{N}$ as our model operates on a sequence of $N$ scalar vector feature tuples $(x_1, f_1), (x_2, f_2), \dots, (x_N, f_N)$ with each feature tuple being $\mathbb{R}^{3} \times \mathbb{R}^{d}$.
> - The notation $F^H$ in Eq. (2) refers to the Hermitian transpose of the FFT matrix.
> - The gating function indeed acts per token.
> - We appreciate the reviewer's focus on clarity and precision of the presentation of the paper! We will fix typos, add clarification and adjust notation accordingly in the revised version.
>
> ## Related work section:
> Due to space constraints, a comprehensive review of related work could not be included in the main text. Instead, we focused on discussing the most essential prior work in the introduction, while providing a more detailed review in Supplementary Section A. This structure is not uncommon with multiple top-tier conference ML papers (e.g., [1,2,3]) adopting a similar format. If the reviewer prefers an explicit related work section in the main text, we are happy to include a shortened version there while retaining the extended related work in the supplementary material.
>
> We thank the reviewer for their constructive feedback! We believe we have addressed all questions and comments. In light of this, we kindly ask the reviewer to consider raising the score.
>
> [1] Gu et al. Mamba: Linear-Time Sequence Modeling with Selective State Spaces. COLM 2024. \
> [2] Yi et al. Bridge the Modality and Capability Gaps in Vision-Language Model Selection. NeurIPS 2024. \
> [3] Razin et a. Implicit Regularization in Deep Learning May Not Be Explainable by Norms. NeurIPS 2020.

---

### Official Review · Reviewer_VW9r · 2025-03-14

**Overall Recommendation:** 3

**Summary:**

The authors introduce a novel equivariant model designed for modeling long geometric sequences. To ensure compatibility with higher-order geometric tensors in global convolution, they propose an equivariant global convolution specifically for vectors. Additionally, they demonstrate that these computations can be performed efficiently using the Fast Fourier Transform (FFT), building on the concept of scalar-valued global convolution. The Geometric Hyena model achieves superior results across various RNA sequencing datasets.

**Claims And Evidence:**

**The Geometric Hyena is designed for tasks that require modeling invariant and equivariant features in geometric graphs.** -- Geometric graphs can appear in different formats. They can be a point cloud with features associated with each point, a mesh, or a big molecule. One issue of using the Hyena or Mamba style model in these cases is converting the geometric graphs into a sequence.
For example, to mamba or state space model for computer vision, researchers explored various ways to convert the $2D$ image into a $1D$ sequence.
However, this case is completely ignored in this case.
The authors might have mostly dealt with RNA data - which, unlike generic geometric graphs, is a definite sequence.

However, unlike RNA seq, most of the geometric graphs do not come with an ordering.

This means the proposed model is not for all "geometric graphs", it is for a small subset of "geometric graphs" where we have some ordering on the nodes. Thus the work do not comply with the main claim.

**Essential References Not Discussed:**

N/A

**Experimental Designs Or Analyses:**

The experiments done are valid but not enough to demonstrate the significance.

**Methods And Evaluation Criteria:**

As claimed to be a generic model for Geometric data, authors should consider all the datasets from EQUIFORMER and VN-Transformer.

Also, as mentioned in line 307 -- "For the other models, we choose hyperparameters so their depth and total parameter count match those of Geometric Hyena, ensuring a fair comparison" --, It is not clear why it is not possible to use the hyper-parameters proposed in the original works.

**Other Comments Or Suggestions:**

1. Type line 241  ”Harry Potter” --> ``Harry Potter''

**Other Strengths And Weaknesses:**

1. The idea of " Geometric associative recall" should be explained further. For example, the statement "In geometric associative recall, each bigram (key and value) is connected by a rotation matrix" is not well understood if not explained in more detail.

**Questions For Authors:**

1. Regarding "Vector long convolution," is this proposed by authors, or is it a standard definition?

**Relation To Broader Scientific Literature:**

The work may have a significant contribution to RNA-Seq analysis. However, as discussed earlier, in general cases, geometric graphs do not have order, thus severely limiting it's application.

**Theoretical Claims:**

N/A

---

> ### Author Rebuttal · Authors · 2025-03-31
>
> ## Application focus of Geometric Hyena:
> Regarding the generalizability of our method to arbitrary geometric graphs, we refer the reviewer to the Limitations section (lines 424–439R), where we explicitly discuss the limitations of G-Hyena for point clouds and emphasize that our method is best suited for problems where a canonical ordering can be established. Also, note that Reviewer YW9h acknowledges in their review that we have noted this limitation explicitly. With this, we disagree that we overclaimed the contribution of our method to arbitrary geometric graphs. That said, we will revise the text to make our target application domain more explicit. Specifically, we will update the phrasing to: "The Geometric Hyena is designed for tasks that require modeling invariant and equivariant features in geometric graphs where canonical order can be established". This will be clarified in the revised version of the paper.
>
> Second, we would like to point out that the class of geometric graphs with canonical ordering is quite broad and includes large bio-molecules such as proteins, antibodies, peptides, enzymes, and RNA - systems that are highly relevant to biosciences and drug discovery. In this context, we also remind the reviewer that our submission is under the Application-Driven Machine Learning track, which explicitly includes biosciences as a focus area in ICML call for papers (icml.cc/Conferences/2025/CallForPapers). In addition, large bio-molecules represent a well-established and rapidly growing application area by its own in the machine learning community, with numerous top-tier papers focused on this domain [1,2,3,4,5]. Viewed in this light, the reviewer’s remark that our work may “have a significant contribution to RNA-Seq analysis” also reflects its broader relevance to machine learning for biosciences. Finally, we note that our method applies beyond RNA, as demonstrated in the protein molecular dynamics task in Section 4.5.
> ## Equiformer and VNT:
> Our method is designed for large bio-molecules, such as RNA or proteins, which can consist of hundreds or thousands of atoms. In contrast, Equiformer and VNT are developed for small geometric systems. For example, the Ribonanza-2k dataset we use has bio-molecules with up to 11300 atoms, whereas the MD17 dataset used to evaluate Equiformer has molecules with only up to 25 atoms. The high dimensionality of our data introduces unique computational challenges that our method is specifically designed to address that do not arise in low-dimensional settings and that existing equivariant models are not equipped to handle at scale. As a result, it is also infeasible to directly reuse the hyperparameters from Equiformer paper (while VNT paper does not report exact optimal hyperparameters). Still following reviewer's suggestion, we tried running Equiformer with the smallest default configuration from the original paper and it runs out of memory on our data even with a batch size of 1 highlighting further the importance of our model for large-scale bio-molecular graphs. Finally, our initial reason for running the models to match the parameter count of G-Hyena was to ensure that we could fairly compare the benefits of these models by isolating the impact of model scale on performance which is a standard practice [6,7].
> ## Associative recall:
> We appreciate the reviewer’s attention to the clarity of presentation of the geometric associative recall task. To aid understanding, we illustrated how “each bigram (key and value) is connected by a rotation matrix” in Figure 4 of the supplementary material. To further improve clarity, we propose updating the caption of Figure 4 to: “A geometric sequence consists of key and value vector tokens, where consecutive key-value pairs form bigrams. Geometric associative recall requires retrieving the value vector corresponding to a query, where the query matches one of the keys in the sequence.”
> ## Vector long convolution:
> Vector long convolution is a novel component introduced in our work that we demonstrate how to implement efficiently in sub-quadratic time. It is not a standard definition.
>
> We thank the reviewer for their constructive feedback! We believe we have addressed all questions and comments. In light of this, we kindly ask the reviewer to consider raising the score.
>
> [1] Groth et al. Kermut: Composite kernel regression for protein variant effects. ICLR 2024.\
> [2] Jing et al. AlphaFold Meets Flow Matching for Generating Protein Ensembles. ICML 2024.\
> [3] Nori et al. RNAFlow: RNA Structure & Sequence Design via Inverse Folding-Based Flow Matching. ICML 2024.\
> [4] Gong et al. Evolution-Inspired Loss Functions for Protein Representation Learning. ICML 2024.\
> [5] Tan et al., Deciphering RNA Secondary Structure Prediction: A Probabilistic K-Rook Matching Perspective. ICML 2024.\
> [6] He et al. Deep Residual Learning for Image Recognition. CVPR 2015.\
> [7] Tan et al. Rethinking Model Scaling for Convolutional Neural Networks. ICML 2019.

---

> > ### Comment · Reviewer_VW9r · 2025-04-05
> >
> > Thanks for the response.
> > Even though the requirement of 'canonical ordering' is mentioned in the limitation, I firmly suggest making it explicit in the introduction and contribution (line 091). Also, make this distinction clear with other equivarinat models in the introduction.
> >
> > For example, in section 2, the author states in line 59 that "A geometric graph of N nodes .... by a **set** of features." Here, I think the authors should have used an ordered set or an index set. Otherwise, the setup seems indistinguishable from the permutation equivarinat setups considered for generic graphs (until the limitation section).
> >
> > While I agree that the proposed method is practical, effective, and scalable for sequence data with geometric features, the write-up and explanation of the paper (in their current state) do not clearly convey this idea.

---

> > > ### Author Response · Authors · 2025-04-05
> > >
> > > Thank you for acknowledging the practicality, effectiveness, and scalability of our method. We appreciate the reviewer's suggestion to emphasize the assumption of canonical ordering more explicitly, and we commit to making it more explicit in the camera-ready version. In addition to the edits in our previous response, we will :
> > >
> > > - Rephrase paragraph 4 in the introduction (051R-063L) "... we propose Geometric Hyena that efficiently models global geometric context in sub-quadratic time while preserving equivariance to rotations and translations. In this work, we focus on a subset of geometric graphs where the canonical order can be established such as biomolecules. The focus on ordered geometric graphs differentiates Geometric Hyena from other equivariant frameworks. Having canonical ordering, we can leverage efficient sequence operators - long convolutions. For equivariance, we introduce vector long convolution that utilizes vector products ... "
> > >
> > > - Rephrase the first contribution to "We propose Geometric Hyena, the first equivariant long-convolutional architecture specifically tailored for large geometric graphs with canonical ordering, designed to efficiently process global geometric context in sub-quadratic time."
> > >
> > > - In 056R-061R, we will clarify: "The Geometric Hyena is designed for tasks that require modeling invariant and equivariant features in geometric graphs where canonical order can be established such as biomolecules."
> > >
> > > - At line 061R, we will add: "The canonical ordering of a geometric graph implies a unique and unambiguous enumeration of its nodes. For instance, in biomolecules, such canonical ordering is naturally established by IUPAC rules [1]. We refer to geometric graphs with canonical ordering as ordered geometric graphs."
> > >
> > > - Instead of the set notation, we will use sequence notation: "An ordered geometric graph of $N$ nodes can be written as a sequence $(x_1, f_1) ... (x_N, f_N)$"
> > >
> > > We believe that these modifications address your concerns, and we are committed to improving the clarity based on your recommendations! With this, we kindly request you to consider these revisions toward increasing your overall score.
> > >
> > > [1] Damhus et al. Nomenclature of inorganic chemistry: IUPAC recommendations. Chem. Int 27, 2005.

---

### Official Review · Reviewer_YW9h · 2025-03-14

**Overall Recommendation:** 5

**Summary:**

This paper presents a novel equivariant SO(3) neural network for processing geometric graphs with sequence structure and invariant node features.  The network a geometric version of Hyena and implements equivariant long convolution in fourier space, allowing for global information flow with subquadratic complexity.  The model is evaluated over a synthetic recall task and several real world RNA property prediction tasks and protein molecular dynamics prediction.

**Claims And Evidence:**

The primary claim of the paper is that geometric hyena can outperform both local equivariant methods and global equivariant attention methods in terms of accuracy and compute time and memory efficiency.    This is well supported both in the design of the method and in the strong empirical results.

**Essential References Not Discussed:**

No

**Experimental Designs Or Analyses:**

- The authors consider a strong set of experiments to test their methods.  The real world RNA and protein experiments demonstrate the method can scale and is practical for interesting real world applications.
- The results are strikingly impressive, with geometric Hyena showing substantial accuracy improvements with much better efficiency and scaling.
- The experiments are carefully chosen to showcase the strength of the method, the inputs have only scalar node features and there is a preferred sequence to use for the points.
- The baselines are well-chosen, representing a selection of well-known, popular, and often SoTA equivariant methods including global attentional methods and local convolutional methods.   These include Equiformer, VectorNeurons Transformer, SchNet, Tensor Field Network, E-GNN, and more.
- I believe non-equivariant methods are included in the recall experiment, but not in the RNA experiments.  It would be nice to see how they compare, for example, with ample data augmentation.  Non-equivariant methods can often perform well too, especially when controlling for memory/compute budget which a focus of the current work.
- Many of the local equivariant methods are targeted more for smaller scale material science tasks, so while they are good to include as baselines, it may not be surprising they do worse on these larger scale tasks with sequence structure.  Is there some way to encode the sequence structure for these models?  Also, although it is not the goal, I wonder how well would Geometric Hyena perform on the types of materials benchmarks those baselines target.

**Methods And Evaluation Criteria:**

- The method builds on Hyena by making it equivariant.  The extension is non-trivial and the method is clearly novel.  For the long convolution layer, elementwise products of scalar are replaced with cross products of vectors.  The authors show this operation can also be performed in the fourier domain in subquadratic time.  vector-scalar interactions are also supported.
- One limitation of this work is that the input can have only scalar node features (and vector positions).  However, the authors have a section in the appendix which extends their framework to higher order SO(3) features and a suggestion on how to implement it efficiently.
- The use of E(N)-GNN for the projection is reasonable since it is fairly efficient and also operators on geometric graphs with scalar node features.  The authors improve E(N)-GNN by adding global context tokens.
- A geometric selective gating mechanism is proposed to emphasize certain tokens, similar to softmax.
- The authors include an ablation study to highlight the importance of the key contributions.  In particular, they compare to the G-transformer which is similar to their method but uses self-attention instead of long convolution.
- A potential limitation of the method, which the author comment directly on, is that long convolution is not permutation equivariant.  Thus for point cloud tasks without a prior sequence structure, a sequence would need to be imposed.  This would likely require significant data augmentation. The authors also note, this could be considered an advantage in tasks like the one they select in which there is a useful sequence structure in the data.
- The normalization of the key and values is considered and implemented to prevent numerical instability.

**Other Comments Or Suggestions:**

- 073R Shouldn’t the input and output spaces for $\Psi$ be the N-fold product of these spaces?
- 111-113R: I found this a touch confusing.  Are they sequences of sets of scalars?  Maybe if the index i is across a sequence, you could use parenthesis instead of curly braces to indicate the order matters.
- Why is the equivariant projection layer called a “projection”?  Isn’t it just a mapping?

**Other Strengths And Weaknesses:**

### Strengths
- Compute and memory efficiency is a large problem under focused on in the equivalence literature.  The focus of this paper, the design of the method, and strong results are very encouraging.  Figure 1 is quite striking.

**Questions For Authors:**

- Is the Fourier transform truncated at some maximum frequency? If so, does it matter where?
- 190L-192L: Does this scalar mapping result in a lot of lost information?
- 220L-223L: Are there any trade-offs with omitting some message pathways this way?
- How big is the sequence length for the protein task?  What is the input feature for the model?

**Relation To Broader Scientific Literature:**

It is a bit unusual to not include a background section or related works section on prior works.  That said, I think the narrative and explanation is clear and that past work and current contributions are clear as written.

**Theoretical Claims:**

The authors include the appropriate theoretical support for their method, proving the equivariance of geometric Hyena in the appendix.

---

> ### Author Rebuttal · Authors · 2025-03-31
>
> We appreciate the reviewer's positive feedback on contributions of our paper, highlighting the novelty ("the extension is non-trivial and the method is clearly novel") of our method and its experimental validation ("the authors consider a strong set of experiments to test their methods").
>
> ## Non-equivariant baselines:
> We agree that the extended comparison with non-equivariant data augmentation baselines will further strengthen the experiments. We train and test Hyena and Transformer models with data augmentation on Open Vaccine, Ribonanza-2k, Tc-ribo and Protein-MD datasets.
>
> |Results (*backbone*) |Open Vaccine|Ribonanza-2k|Tc-ribo|Protein-MD|
> |-----|------------------------|------------------------|-------------------|----------------|
> |Hyena|0.447±0.037|0.810±0.124|0.560±0.002|48.94±9.03|
> |Transformer|0.400±0.004|0.637±0.006|0.556±0.001|75.83±6.35|
> |G-Hyena|**0.363±0.045**|**0.529±0.005**|**0.517±0.025**|**1.80±0.009**|
>
> |Results (*all-atom*)|Open Vaccine|Ribonanza-2k|Tc-ribo|Protein-MD|
> |-----|------------------------|------------------------|-------------------|----------------|
> |Hyena|0.393±0.002|0.605±0.017|0.569±0.001|55.34±5.51|
> |Transformer|0.399±0.004|0.633±0.007|**0.553±0.002**|79.68±24.2|
> |G-Hyena|**0.339±0.004**|**0.546±0.006**|**0.552±0.003**|**2.49±0.037**|
>
> Results suggest that on Open Vaccine and Ribonanza-2k data, non-equivariant models lag significantly behind Geometric Hyena (with G-Hyena delivering up to 34% error reduction), and on Tc-Ribo all-atom Transformer performs on par with. Yet, non-equivariant models struggle to generalize on the Protein-MD task where non-equivariant Hyena and Transformer perform poorly compared to Geometric Hyena.
>
> ## Local methods with sequence-structured priors:
> Extending local methods with sequence-structured prior presents a non-trivial direction for future work that extends beyond the scope of our paper. However, to foster future research, we conducted an extra experiment where we employed a Hyena layer to extract sequence features and then ran EGNN on top (Seq-EGNN). Alternatively, we can first run EGNN to extract geometric features and then run a sequence model on top (EGNN-Seq).
>
> |Results|Open Vaccine (*backbone*)|Open Vaccine (*all-atom*)|
> |-----|-------------------------------|--------------------------------|
> |EGNN|0.529±0.006|0.511±0.005|
> |Seq-EGNN|0.527±0.006|0.506±0.009|
> |EGNN-Seq|0.489±0.002|0.490±0.004|
> |G-Hyena|**0.363±0.045**|**0.339±0.004**|
>
> Our initial results suggest that adding sequence prior has little effect when EGNN is applied on top of sequence features. However, when the sequence model is applied on top of EGNN features, we observe slight improvement, suggesting potential for further research in this direction.
>
> ## Future benchmarking on materials:
> In our work, we focus specifically on large bio-molecules. We agree that extending experimental comparison to a broader domain, including materials, presents an important direction for future work, and we plan to do so in the future. We would appreciate if the reviewer could point us to relevant material benchmarks that include large molecules for our future work.
>
> ## Higher-order Geometric Hyena:
> Recent works [1,2] observed that higher-order representations provide diminishing performance improvement while significantly increasing computational requirements and memory footprint. This is also supported by the excellent performance of scalarization-based equivariant GNNs [3]. Based on this evidence, we decided not to proceed with a higher-order version of Geometric Hyena since our focus is on computational and memory efficiency.
>
> ## Other comments and questions:
> - Indeed, our model operates on a sequence of $N$ scalar vector feature tuples $(x_1, f_1), (x_2, f_2), \dots, (x_N, f_N)$ rather than unordered set, we thank the reviewer for pointing this out. With this, a more precise formulation of the domain and co-domain would be $\Psi: (\mathbb{R}^{3} \times \mathbb{R}^{d})^{N} \rightarrow (\mathbb{R}^{3} \times \mathbb{R}^{d})^{N}$.
> - Projection terminology is used for alignment with the terminology in the Transformer and Hyena papers.
> - In our implementation, we use standard discrete FFT with circular convolution. There is no explicit frequency truncation and all available frequencies are used up to the Nyquist limit.
> - 190-192L: Yes, similar to the scalarization trick [3], this results in a lossy compression of information.
> - In Protein-MD each protein has 855 and 3341 atoms in backbone and all-atom version. Atom identities are used as feature for all methods.
> - All clarifications, discussion, and new results will be added to the revised version of the paper.
>
> [1] Brandstetter et al. Geometric and Physical Quantities Improve E(3) Equivariant Message Passing. ICLR 2022. \
> [2] Wand et al. Rethinking the Benefits of Steerable Features in 3D Equivariant Graph Neural Networks. ICLR 2024. \
> [3] Satorras et al. E(n) Equivariant Graph Neural Networks. ICML 2021.

---

> > ### Comment · Reviewer_YW9h · 2025-04-04
> >
> > I've read the other reviews and responses. I don't really see anything that changes my opinion. I appreciate the answers to my questions and non-equivariant baselines.

---

### Decision · Program_Chairs · 2025-05-01

**Decision:**

Accept (spotlight poster)

**Comment:**

This paper introduces "Geometric Hyena Networks", which is a method for applying long-convolutions while preserving Euclidan equivariance. This method leads to improved results in terms of accuracy and speed in comparison with standard equivariant methods. The reviewers were mostly supportive of the paper, and hence I  recommend accepting the paper.